# Asymptotically Optimal Sequential Testing with Markovian Data

**Alhad Sethi**[1]   **Kavali Sofia Sagar**[2]   **Shubhada Agrawal**[1]   **Debabrota Basu**[3]   **P. N. Karthik**[2]

## Abstract

We study one-sided and $\alpha$-correct sequential hypothesis testing for data generated by an ergodic, finite-state Markov chain. The *null* hypothesis is that the unknown transition matrix belongs to a prescribed set $\mathcal{P}$ of stochastic matrices, and the *alternative* corresponds to a disjoint set $\mathcal{Q}$. We establish *a non-asymptotic instance-dependent lower bound* on the expected stopping time of any valid sequential test under the alternative, which is asymptotically tight. Our novel analysis improves the existing lower bounds, which are either asymptotic or provably sub-optimal in this setting. Our lower bound incorporates both the stationary distribution and the transition structure induced by the unknown Markov chain. We further propose an optimal test whose expected stopping time matches this lower bound asymptotically as $\alpha \to 0$. We illustrate the usefulness of our framework through applications to sequential detection of model misspecification in Markov Chain Monte Carlo and to testing structural properties, such as the linearity of transition dynamics, in Markov decision processes. Our findings yield a sharp and general characterization of optimal sequential testing procedures under Markovian dependence.

## 1. Introduction

Hypothesis testing is a cornerstone of statistics and theoretical computer science: from data, one decides whether an unknown data-generating mechanism satisfies a prescribed property (Lehmann & Romano, 2005; Goldreich, 2017). Classical theory largely assumes i.i.d. samples, but many modern streams are temporally dependent, making hypothesis testing under dependence both practically important and theoretically subtle (Phatarfod, 1965; Gyori & Paulin, 2015; Fauß et al., 2020).

A widely used model of dependence is *Markovianity*, where the future is conditionally independent of the past given the present state (Bengio et al., 1999; Nagaraj et al., 2020). Markov dynamics arise in Markov Chain Monte Carlo (MCMC) (Roy, 2020), reinforcement learning and control (Sutton et al., 1998; Bertsekas, 2017; Beutler & Ross, 1985), and hidden Markov models (Rabiner & Juang, 2003). In these settings, the transition mechanism is typically unknown; theory often proceeds by imposing structural assumptions (e.g., Gaussianity or bilinearity) (Jin et al., 2020; Ouhamma et al., 2023), whose validity may be unclear. This motivates *testing* model classes under Markovian data (Natarajan, 2003; Fauß et al., 2020).

We study *sequential* hypothesis testing for finite-state Markov chains. Let $[m] := \{1, \ldots, m\}$, $\Delta_m$ be the simplex in $\mathbb{R}^m$, and $\mathcal{M}$ the set of $m \times m$ row-stochastic matrices. A time-homogeneous Markov chain is specified by $(P, \mu)$ with $P \in \mathcal{M}$ and $\mu \in \Delta_m$, generating $X_1, X_2, \ldots$ via

$$\mathbb{P}_{(P,\mu)}[X_1 = x_1, \ldots, X_n = x_n] = \mu(x_1) \prod_{t=2}^{n} P(x_{t-1}, x_t).$$

We assume *sequential access* to data: at time $t$ we observe $X_t$. Given an unknown $(P, \mu)$, we test whether $P$ lies in a prescribed null class $\mathcal{P} \subset \mathcal{M}$ versus an alternative $\mathcal{Q} \subset \mathcal{M}$:

$$H_0: \ P \in \mathcal{P} \qquad \text{versus} \qquad H_1: \ P \in \mathcal{Q},$$

a *composite versus composite* problem. We assume $\mathcal{P} \cap \mathcal{Q} = \varnothing$ (and impose additional separation/identifiability conditions only when required for sharp characterizations).

We work in the one-sided, $\alpha$-correct, power-one sequential framework (Darling & Robbins, 1967; Farrell, 1964; Robbins & Siegmund, 1974). For $\alpha \in (0, 1]$, an $\alpha$-*correct, power-one sequential test* is a stopping time $\tau_\alpha$ (rejecting $H_0$ upon stopping) such that, uniformly over $\mu \in \Delta_m$,

$$\mathbb{P}_{P,\mu}[\tau_\alpha < \infty] \leq \alpha \qquad \forall P \in \mathcal{P},$$
$$\mathbb{P}_{Q,\mu}[\tau_\alpha < \infty] = 1 \qquad \forall Q \in \mathcal{Q}.$$

Suppressing $\mu$ for brevity, our objective is *instance-dependent* efficiency under the alternative:

---

[1]Department of Electrical Communication Engineering, Indian Institute of Science, Bangalore, India [2]Department of Artificial Intelligence, Indian Institute of Technology, Hyderabad, India [3]Univ. Lille, Inria, CNRS, Centrale Lille, UMR 9189 – CRIStAL, Lille, France. Correspondence to: Alhad Sethi <alhadsethi@iisc.ac.in>.

*Proceedings of the 43rd International Conference on Machine Learning*, Seoul, South Korea. PMLR 306, 2026. Copyright 2026 by the author(s).

1. For fixed $Q \in \mathcal{Q}$, what is $\inf \mathbb{E}_Q [\tau_\alpha]$ over all $\alpha$-correct, power-one tests?
2. Can we design a procedure that achieves this infimum *to first order* as $\alpha \to 0$, simultaneously for all $Q \in \mathcal{Q}$?

## 1.1. Contributions

Our main contributions are as follows.

1. **Non-asymptotic instance-dependent lower bound and asymptotically optimal test.** We prove the first *non-asymptotic, instance-dependent* lower bound on $\mathbb{E}_Q [\tau_\alpha]$ for $\alpha$-correct, power-one sequential tests under an *unknown* alternative $Q \in \mathcal{Q}$ (Theorem 3.3). The leading term is $\log(1/\alpha)$ scaled by an information quantity $D_{\mathcal{M}}^{\inf}(Q, \mathcal{P})$, an infimum of stationary-weighted KL divergences over $\mathcal{P}$, plus an $\alpha$-independent term depending on structural properties of $Q$. We then construct a sequential test that matches this bound *to first order* as $\alpha \to 0$ (Theorem 4.1).

2. **Composite null is fundamentally harder than a known singleton null.** Fields et al. (2025) study *singleton null versus composite alternative*: $H_0 : P = P_0$ for known $P_0$, versus $H_1 : P \in \mathcal{Q}$. Our setting is *composite versus composite* with a *composite* null: even when data is generated by a specific $Q$, the test must rule out *every* $P \in \mathcal{P}$ while maintaining uniform Type-I control. This necessity of certifying incompatibility with an entire null class (rather than a single known reference) drives both our information characterization (via an infimum over $\mathcal{P}$) and the technical analysis.

3. **Two technical tools.** We state and prove two results that may be of independent interest: (a) A Pinsker-type inequality (Proposition 4.3) lower bounding a stationary-weighted divergence between two transition matrices by squared gaps between stationary means of suitable functions. (b) A uniform control of solutions to the Poisson equation in terms of mixing properties (Proposition 3.1).

4. **Applications.** We instantiate our framework for (i) *misspecification detection* in MCMC by testing consistency with a target stationary distribution (Section 5.1), obtaining optimal detection guarantees (Corollary 5.1); and (ii) *structural testing* in RL by sequentially verifying linear transition dynamics in MDPs (Section 5.2).

## 1.2. Related Works

**Markovian Sequential Testing.** Sequential testing dates back to Wald (1945) and the SPRT for two simple hypotheses under i.i.d. data. We refer the reader to Agrawal & Ramdas (2025, Section 2) for an extensive survey on sequential testing under i.i.d. data. Work on sequential testing under Markovian dependence is comparatively limited, but includes early contributions such as Phatarfod (1965;

1971); Schmitz & Süselbeck (1983); Dimitriadis & Kazakos (2007); Kiefer & Sistla (2016), which are largely SPRT-type procedures tailored to *simple* hypotheses (singleton null and singleton alternative).

Fauß et al. (2020) study sequential and fixed-sample testing for Markov processes from a minimax/robust perspective, deriving tests with optimal worst-case guarantees. Their objective is complementary to ours: we consider one-sided, $\alpha$-correct, power-one tests and seek *instance-dependent* characterizations under the (unknown) alternative.

The closest work to ours is Fields et al. (2025), which analyzes one-sided sequential tests for Markov chains based on the plug-in likelihood estimator of Takeuchi et al. (2013). Their setting differs in two key respects: (i) they test a *singleton* null against its complement, whereas we treat *composite* null and *composite* alternative classes; and (ii) their analysis assumes uniformly bounded likelihood ratios, which we do not require. Moreover, the above works do not provide *instance-dependent lower bounds* on the expected stopping time. In contrast, we establish a non-asymptotic, instance-dependent lower bound and match it (to first order) with an explicit procedure, proving asymptotic optimality without boundedness assumptions.

**Multi-armed bandits and RL.** Best arm identification in Markovian bandits (Anantharam et al., 2003; Moulos, 2019; Karthik et al., 2024) is closely related: one-sided sequential testing can be viewed as a *single-armed* fixed-confidence identification problem, where one observes a single evolving process and stops with controlled error once sufficient evidence accumulates.

Sharp fixed-confidence results for Markovian bandits, however, typically rely on strong parametric structure, most notably *single-parameter exponential family* (SPEF) assumptions on the transition models (or reward/transition parametrizations) (Anantharam et al., 2003; Moulos, 2019; Karthik et al., 2024). Even in such regimes, lower bounds are often asymptotic and achievability arguments exploit the parametric form. While our setting admits a single-process interpretation, *we do not require an SPEF assumption*: we derive non-asymptotic, instance-dependent lower bounds and matching (first-order) achievability for substantially more general Markov dynamics.

A related but distinct line concerns *best policy identification* (BPI) in MDPs (Al Marjani & Proutiere, 2021; Al Marjani et al., 2021; Wagenmaker et al., 2022; Tuynman et al., 2024), which provides non-asymptotic lower bounds for identifying an optimal policy under active data collection. Directly adapting these bounds to our testing problem can be loose; see Section A.4. Nevertheless, we leverage martingale constructions from this literature in our test design and analysis.

Closest in spirit to our work is *policy testing* in MDPs. Ariu et al. (2025) derive asymptotic lower bounds and propose procedures that match them asymptotically using martingale techniques. Although BPI-style non-asymptotic lower bounds apply in principle, they need not be tight for policy testing. Our instance-dependent, non-asymptotic methodology can be adapted to obtain sharper lower bounds in this setting as well, beyond parametric SPEF-type regimes.

**Paper organization:** Section 2 introduces background and notation. Section 3 establishes the instance-dependent lower bound. Section 4 presents an asymptotically optimal test with matching sample-complexity guarantees. Section 5 contains applications: Section 5.1 (MCMC misspecification testing) and Section 5.2 (structural testing in MDPs). Proofs are deferred to the appendices.

## 2. Preliminaries: Primer on Markov Chains

This section collects the notations and standard background on finite-state Markov chains used throughout the paper.

**Notation.** We denote the natural numbers, real numbers, and strictly positive real numbers by $\mathbb{N}$, $\mathbb{R}$, and $\mathbb{R}_{++}$, respectively. For $m \in \mathbb{N}$, let $[m] := \{1, \ldots, m\}$. We write $\mathbf{1}_{n \times m}$ for the all-ones matrix (dropping subscripts when dimensions are clear). Any function $f : [m] \to \mathbb{R}$ is identified with the vector $f \in \mathbb{R}^m$, and we use these representations interchangeably. For $x \in \mathbb{R}$, define $x^+ := \max\{x, 0\}$.

Let $\Delta_m$ be the probability simplex in $\mathbb{R}^m$. For $p, q \in \Delta_m$, we write $p \ll q$ if $q(i) = 0$ implies $p(i) = 0$ for all $i \in [m]$. For a random variable $X$, $\sigma(X)$ denotes the $\sigma$-algebra it generates. For $p \in \Delta_m$ and $f : [m] \to \mathbb{R}$, define $\mathbb{E}_p[f] := \sum_{i \in [m]} f(i)p(i)$, and let $\mathbb{I}$ denote the indicator function.

**Transition matrices and norms.** A transition matrix on $m$ states is a nonnegative row-stochastic matrix $P \in \mathbb{R}^{m \times m}$ satisfying $P(i, j) \geq 0$ for all $i, j \in [m]$, and $P\mathbf{1} = \mathbf{1}$. Let $\mathcal{M}$ denote the set of all such matrices, and write $P(i, \cdot) \in \Delta_m$ for the $i$th row of $P$. For $P, Q \in \mathcal{M}$, we say $Q$ is (row-wise) absolutely continuous with respect to $P$, denoted $Q \ll P$, if $Q(i, \cdot) \ll P(i, \cdot)$ for all $i \in [m]$. We use $\mathbb{E}_Q[\cdot]$ to denote expectations when the chain evolves according to transition matrix $Q$ (with the initial distribution clear from context). We equip $\mathbb{R}^{m \times m}$ with the $\|\cdot\|_{1,\infty}$ norm (Wolfer & Kontorovich, 2019), defined as

$$\|A\|_{1,\infty} := \max_{i \in [m]} \|A(i, \cdot)\|_1, \qquad A \in \mathbb{R}^{m \times m}.$$

For $A \in \mathcal{M}$, one has $\|A\|_{1,\infty} = 1$, but the norm is most useful through the metric it induces on $\mathcal{M}$:

$$\|P - Q\|_{1,\infty} = 2 \max_{i \in [m]} \|P(i, \cdot) - Q(i, \cdot)\|_{\mathrm{TV}},$$

i.e., twice the worst-case total variation distance between corresponding rows.

**Ergodic Markov chains.** Let $(X_t)_{t \geq 0}$ be a Markov chain on $[m]$ with transition matrix $P \in \mathcal{M}$ and initial distribution $\mu^\top \in \Delta_m$. A distribution $\pi^\top \in \Delta_m$ is *stationary* for $P$ if $\pi P = \pi$. We call $P$ (and the induced chain) *ergodic* if it is irreducible and aperiodic. In this case, $P$ admits a unique stationary distribution $\pi^\top$ with $\pi_* := \min_{i \in [m]} \pi(i) > 0$, and $\pi$ is also the limiting visitation distribution: $\lim_{n \to \infty} \|\mu P^n - \pi\|_{\mathrm{TV}} = 0$.

**Assumption 2.1.** The data-generating Markov chain is ergodic.

The above assumption is not too restrictive, as the set of ergodic transition matrices is dense in $\mathcal{M}$. Concretely, for any $P \in \mathcal{M}$ and $\epsilon > 0$, the matrix $P_\epsilon := \left(1 - \frac{\epsilon}{m}\right)P + \left(\frac{\epsilon}{m}\right)\mathbf{1}_{m \times m}$ is ergodic and can be made arbitrarily close to $P$ as $\epsilon \to 0$; see Levin et al. (2017, Chapter 1).

**Poisson equation.** A central tool in Markov chain analysis is the Poisson equation (PE). For an ergodic $P \in \mathcal{M}$ with stationary distribution $\pi^\top$ and a function $f : [m] \to \mathbb{R}$, the PE for $(P, f)$ (in the unknown $\omega$) is

$$(I - P)\omega = f - (\pi f)\mathbf{1}, \tag{1}$$

where $\pi f := \mathbb{E}_\pi[f]$ is a scalar and $\mathbf{1}$ is the all-ones column vector. The PE (1) always admits solutions; one convenient choice is

$$\omega_{P,f} := \sum_{n=0}^{\infty}(P^n - \mathbf{1}\pi)\, f, \tag{2}$$

where $\mathbf{1}\pi$ denotes the rank-one matrix with every row equal to $\pi$. We refer to Douc et al. (2018, Section 21.2) for further background. The PE is useful because it enables a standard decomposition of additive functionals into a martingale difference term plus a remainder that is typically negligible under mixing.

**Spectral properties.** Spectral information about $P$ quantifies its long-run behavior and mixing. Fix an ergodic $P \in \mathcal{M}$ with stationary distribution $\pi^\top$. The chain is *reversible* if it satisfies detailed balance: $\pi(i)P(i, j) = \pi(j)P(j, i)$ for all $i, j \in [m]$. For reversible $P$, all eigenvalues lie in $[-1, 1]$ and can be ordered as $1 = \lambda_1 \geq \lambda_2 \geq \cdots \geq \lambda_m$ (Levin et al., 2017, Lemma 12.2). The (usual) spectral gap is then $\gamma(P) := 1 - \lambda_2$. For non-reversible chains, several notions of spectral gap exist (Fill, 1991; Paulin, 2015; Chatterjee, 2025). We follow Paulin (2015) and work with the *pseudo-spectral gap*.

**Definition 2.2** (Pseudo-spectral gap, Paulin (2015))**.** For an ergodic $P \in \mathcal{M}$ with stationary distribution $\pi^\top$, let $P^*$ be its time-reversal, defined via $P^*(i, j) := P(j, i)\frac{\pi(j)}{\pi(i)}$. The pseudo-spectral gap of $P$ is

$$\gamma_{\mathrm{ps}}(P) := \max_{k \geq 1} \frac{\gamma\big((P^*)^k P^k\big)}{k}.$$

One has $\gamma_{\mathrm{ps}}(P) \in (0, 1]$ for ergodic $P$, and larger $\gamma_{\mathrm{ps}}(P)$ corresponds to faster mixing (Paulin, 2015, Proposition 3.4). In our development, $\gamma_{\mathrm{ps}}$ enters the lower-bound analysis; the algorithm itself is largely insensitive to this particular choice (see Remark A.5).

## 3. Instance-dependent Lower Bound

In this section, we derive a non-asymptotic lower bound on the expected stopping time of any $\alpha$-correct, power-one sequential test for Markov chains. The bound holds for every $\alpha \in (0, 1)$. We first introduce two quantities that will appear in the statement.

**Proposition 3.1.** *For an ergodic $P \in \mathcal{M}$ and $f \in \mathbb{R}^m$, let $\omega_{P,f} \in \mathbb{R}^m$ be the solution (2) to the PE for $(P, f)$. Then*

$$\|\omega_{P,f}\|_\infty \leq C_P \|f\|_\infty,$$

*where, writing $\gamma_{\mathrm{ps}} = \gamma_{\mathrm{ps}}(P)$, the constant $C_P$ depends only on $P$ and is given by*

$$C_P := \begin{cases} \frac{(1-\gamma_{\mathrm{ps}})^{-\frac{1}{2\gamma_{\mathrm{ps}}}}}{\sqrt{\pi_*}} \left( \frac{1}{1-\sqrt{1-\gamma_{\mathrm{ps}}}} \right), & \gamma_{\mathrm{ps}} \in (0, 1), \\ 2, & \gamma_{\mathrm{ps}} = 1. \end{cases}$$

*Interpretation.* The quantity $C_P$ controls the *sensitivity* of the Poisson solution: it upper bounds the magnitude of $\omega_{P,f}$ relative to that of $f$, in terms of the mixing properties of $P$ (captured by $\gamma_{\mathrm{ps}}$ and $\pi_*$).

When $\gamma_{\mathrm{ps}} = 1$, the chain behaves as in the i.i.d. case: $P$ has identical rows equal to its stationary distribution, so $P^n = \mathbf{1}\pi$ for all $n \geq 1$. Substituting into (2) yields $\omega_{P,f} = f - (\pi f)\mathbf{1}$ and hence $C_P = 2$. For $\gamma_{\mathrm{ps}} \in (0, 1)$, the proof combines the representation (2) with bounds on $\|P^n - \mathbf{1}\pi\|$ in terms of $\gamma_{\mathrm{ps}}$ from Paulin (2015). The proof is given in Section A.2.

Next we define the information-theoretic quantity that governs the hardness of testing.

**Definition 3.2** (Stationary-weighted KL divergence)**.** Let $P, Q \in \mathcal{M}$ be such that $Q \ll P$ and $Q$ is ergodic, and let $\pi^\top \in \mathbb{R}^m$ be the stationary distribution of $Q$. Let

$$D_{\mathcal{M}}(Q, P) := \sum_{i \in [m]} \pi_i \, D_{\mathrm{KL}}\left(Q(i, \cdot), P(i, \cdot)\right),$$

where for $q, p \in \Delta_m$, $D_{\mathrm{KL}}(q, p) := \sum_{j \in [m]} q_j \log \frac{q_j}{p_j}$.

With the above notations in place, we now state the first main result of our paper: instance-dependent lower bound on the expected stopping time.

**Theorem 3.3** (Lower bound)**.** *Fix $\alpha \in (0, 1)$ and an ergodic $Q \in \mathcal{Q}$ with stationary distribution $\pi^\top \in \mathbb{R}^m$. Let $\tau_\alpha$ be the stopping time of any $\alpha$-correct, power-one sequential test. Then for every $P \in \mathcal{P}$,*

$$\mathbb{E}_Q\left[\tau_\alpha\right] \geq \frac{\log(1/\alpha)}{D_{\mathcal{M}}(Q, P)} - \frac{\mathbb{E}_Q\left[\omega_{Q,f_P}(X_0) - \omega_{Q,f_P}(X_{\tau_\alpha})\right]}{D_{\mathcal{M}}(Q, P)},$$

*where $f_P(i) := D_{\mathrm{KL}}\left(Q(i, \cdot), P(i, \cdot)\right)$ for all $i \in [m]$, and $\omega_{Q,f_P}$ denotes the solution (2) for $(Q, f_P)$. Moreover,*

$$\mathbb{E}_Q\left[\tau_\alpha\right] \geq \left( \frac{\log(1/\alpha)}{D_{\mathcal{M}}^{\mathrm{inf}}(Q, \mathcal{P})} - \frac{2C_Q}{\min_i \pi_i} \right)^+, \qquad (3)$$

*where $D_{\mathcal{M}}^{\mathrm{inf}}(Q, \mathcal{P}) := \inf_{P \in \mathcal{P}} D_{\mathcal{M}}(Q, P)$ and $C_Q$ is the constant in Proposition 3.1.*

**Proof sketch.** We apply a Markov-chain version of Wald's lemma (Moustakides, 1999) to express the expected log-likelihood ratio between $Q$ and $P$ as $\mathbb{E}_Q\left[\tau_\alpha\right] D_{\mathcal{M}}(Q, P)$ plus an additive term involving a Poisson solution for $(Q, f_P)$. We then invoke the data processing inequality to relate the log-likelihood ratio to the test error, which yields the first inequality. To obtain (3), we optimize over $P \in \mathcal{P}$ and upper bound the Poisson correction using Proposition 3.1. Full details appear in Section A.

Taking $\alpha \to 0$ in (3) yields the asymptotic relation

$$\liminf_{\alpha \to 0} \frac{\mathbb{E}_Q\left[\tau_\alpha\right]}{\log(1/\alpha)} \geq \frac{1}{D_{\mathcal{M}}^{\mathrm{inf}}(Q, \mathcal{P})}$$

for every $\alpha$-correct, power-one test. In particular, $D_{\mathcal{M}}^{\mathrm{inf}}(Q, \mathcal{P})$ is the *fundamental instance-dependent hardness* of certifying that the data-generating transition matrix is $Q$ (against $\mathcal{P}$).

*(a) Relation to non-asymptotic RL lower bounds.* Non-asymptotic lower bounds in best policy identification for MDPs (Al Marjani & Proutiere, 2021; Al Marjani et al., 2021; Wagenmaker et al., 2022; Tuynman et al., 2024) often proceed by (i) writing the expected log-likelihood ratio as a weighted sum of per-state divergences $f_P(i)$, (ii) applying data processing, and (iii) optimizing over the weights. This yields denominators of the form $\sup_{w \in \Delta_m} \inf_{P \in \mathcal{P}} w^\top f_P$. While valid, this relaxation can be loose in our setting because we do *not* control state visitation: the weights are fixed by the instance $Q$ through its stationary distribution. In particular, even in the simple case $\mathcal{P} = \{P\}$, the optimizer over $w$ places all mass on the state with the largest KL term, which is not achievable when the visitation proportions are dictated by $Q$.

*(b) Connection to the i.i.d. case.* In the i.i.d. setting, Agrawal & Ramdas (2025) characterize hardness via the KL projection of $Q$ onto $\mathcal{P}$. Theorem 3.3 shows that the analogous quantity in the Markovian setting is the stationary-weighted projection $D_{\mathcal{M}}^{\mathrm{inf}}(Q, \mathcal{P})$. A key technical difference is that the

Poisson correction term depends on $P$ through $f_P$. Therefore, naively taking a supremum over $P$ (as one can in the i.i.d. analysis) can make the correction term arbitrarily large and destroy the bound. We avoid this by controlling the correction uniformly using Proposition 3.1, which depends only on the alternative $Q$.

*Remark* 3.4 (Recovery of i.i.d. bounds). Consider the following testing problem: given i.i.d. samples from some distribution in $\Delta_m$, we want to test whether the samples are drawn from $q \in \Delta_m$ or from some distribution in a disjoint set $\mathcal{P}_{\mathrm{iid}} \subset \Delta_m$. This is a special case of the Markovian case and corresponds to the setting where $Q \in \mathcal{M}$ has identical rows equal to $q$, and $\mathcal{P} \subset \mathcal{M}$ consists of matrices with identical rows equal to some $p \in \mathcal{P}_{\mathrm{iid}}$. In this case, the per-state divergence vector $f_P$ (defined in Theorem 3.3) is constant. Consequently, the zero vector satisfies the Poisson equation for $(Q, f_P)$, allowing us to take $C_Q = 0$. We thus recover the standard i.i.d. bounds (Agrawal & Ramdas, 2025, Theorem 3.1).

*(c) Implications for best arm identification in Markovian bandits.* A one-sided sequential test can be viewed as a single-armed Markovian bandit instance (Anantharam et al., 2003; Moulos, 2019; Karthik et al., 2024). Existing lower bounds in this literature are typically asymptotic (e.g., $\alpha \to 0$) and/or rely on single-parameter exponential family (SPEF) assumptions. Our proof technique suggests a route to non-asymptotic instance-dependent lower bounds without the SPEF assumption.

*(d) Implications for policy testing.* In policy testing, one collects Markovian trajectories under a fixed policy and tests whether the expected cumulative reward exceeds a threshold. Our argument can be adapted to yield non-asymptotic lower bounds for this problem, complementing the currently available asymptotic results (Ariu et al., 2025).

## 4. Algorithm Design and Optimality

We now present a sequential procedure for testing a compact null hypothesis set $\mathcal{P}$ against an alternative hypothesis set $\mathcal{Q}$, with $\mathcal{P} \cap \mathcal{Q} = \varnothing$. Unlike much of the prior literature—which typically assumes at least one of $\mathcal{P}, \mathcal{Q}$ is a singleton (Phatarfod, 1965; Fields et al., 2025)—our framework accommodates *composite* structure for both sets.

Our method, outlined in Algorithm 1, estimates the transition matrix from observed data and employs a martingale-based test statistic. In brief, it constructs an empirical transition kernel $\widehat{Q}_t$ from state-transition counts, computes the statistic $L_t$ (Line 16), and stops once $L_t$ exceeds a state-visitation-count-dependent boundary $\beta_t$ (Line 17), at which point it rejects the null. We establish the asymptotic optimality of Algorithm 1.

**Theorem 4.1.** *For any $\alpha \in (0, 1)$, the test (Algorithm 1) is*

---

**Algorithm 1** Sequential Markov Chain Test

**Require:** State space $[m]$, (null) set $\mathcal{P}$, $\alpha \in (0, 1)$.
1: **Initialize:** $t \leftarrow 0$, observe initial state $X_0$.
2: **Initialize Counts:** $N_x \leftarrow 0$, $N_{x,y} \leftarrow 0 \ \forall \, x, y \in [m]$.
3: **loop**
4:    $t \leftarrow t + 1$
5:    **Observe** next state $X_t$.
6:    $u \leftarrow X_{t-1}, \ v \leftarrow X_t, \ N_u \leftarrow N_u + 1, \ N_{u,v} \leftarrow N_{u,v} + 1$
7:    **for** $x \in [m]$ **do**
8:      **if** $N_x > 0$ **then**
9:        $\widehat{Q}_t(x, \cdot) \leftarrow [N_{x,1}/N_x, \ldots, N_{x,m}/N_x]$
10:      **else**
11:        $\widehat{Q}_t(x, \cdot) \leftarrow [1/m, \ldots, 1/m]$
12:      **end if**
13:    **end for**
14:    $\psi_t \leftarrow \sum_{x \in [m]} \log\left( e \left( 1 + \frac{N_x}{m-1} \right) \right)$
15:    $\beta_t \leftarrow \log(1/\alpha) + (m-1)\psi_t$
16:    $L_t \leftarrow \inf_{P \in \mathcal{P}} \sum_{x : N_x > 0} N_x \, \mathrm{D_{KL}}\left( \widehat{Q}_t(x, \cdot), P(x, \cdot) \right)$
17:    **if** $L_t \geq \beta_t$ **then**
18:      **Stop** (Reject $\mathcal{P}$) and set $\tau_\alpha = t$
19:    **end if**
20: **end loop**

---

$\alpha$-*correct. Moreover, for any $Q \in \mathcal{Q}$,*

$$\limsup_{\alpha \to 0} \frac{\mathbb{E}_Q[\tau_\alpha]}{\log(1/\alpha)} \leq \frac{1}{D_{\mathcal{M}}^{\inf}(Q, \mathcal{P})}. \tag{4}$$

We omit the proof here for brevity and refer the reader to Section B for a detailed proof.

*Remark* 4.2. Note that the (non-negative) process $M_t := e^{L_t - (m-1)\psi_t}$ is upper bounded by a non-negative supermartingale (see, Appendix B.3, (22)), and hence, is an e-process. We refer the reader to cf. Ramdas & Wang (2025, §7) for a definition of an e-process.

### 4.1. Computational Tractability

Since the test statistic $L_t$ optimizes a convex objective over $P \in \mathcal{P}$, when $\mathcal{P}$ is convex we can compute $L_t$ efficiently using standard convex optimization tools. In Section 5, we illustrate this in two instances—testing misspecification of MCMC samplers and testing linearity of transitions in Markov Decision Processes (MDPs)—where the null set is convex and the statistic can be evaluated efficiently.

In general, our setup allows the null set $\mathcal{P}$ to be arbitrarily non-convex. Consequently, designing computationally efficient implementations requires exploiting additional structure in $\mathcal{P}$ (cf. Al Marjani & Proutiere (2021); Ariu et al. (2025)). When $\mathcal{P}$ is a finite union of convex sets, one can solve the convex program over each component and then take the minimum (Carlsson et al., 2024; Das et al., 2025).

For computational efficiency, recent work has also explored Transformer-based surrogates for sequential testing/pure-exploration style problems (Russo et al., 2026). While promising empirically, such approaches do not, in general, come with (a) rigorous, user-prescribed $\alpha$-type guarantees of the kind required here, and (b) a clear mechanism that remains well-calibrated and scalable in the fixed-confidence regime $\alpha \to 0$. We address both issues in one step via the next proposition, which provides a principled Pinsker-type *lower bound* on our statistic: thresholding this lower bound preserves $\alpha$-correctness by construction, while yielding a computationally tractable alternative. The following result may be of independent interest.

**Proposition 4.3.** *Let $P, Q \in \mathcal{M}$ be ergodic with stationary distributions $\pi_P, \pi_Q$, respectively. For any $g : [m] \to \mathbb{R}$ not constant over states,*

$$D_{\mathcal{M}}(Q, P) \geq \frac{1}{2\|\omega_{P,g}\|_\infty^2} \left( \mathbb{E}_{\pi_Q}[g] - \mathbb{E}_{\pi_P}[g] \right)^2, \quad (5)$$

*where $\omega_{P,g}$ is a solution to the PE for the pair $(P, g)$.*

Recall that in case $g$ is constant, the difference in expectations is zero and we get $0/0$ which we take by convention to be $0$. Hence, the bound is trivial. A similar approximation (lower bound) for the corresponding complexity term in the i.i.d. setting was proven in Agrawal et al. (2021a, Section 3.4). However, proving it in the Markovian setting is substantially more delicate and requires new ideas. The main technical challenge is to pass from a difference in stationary expectations to a sum of row-wise divergences. We overcome this by combining the Poisson equation with the variational characterization of total variation distance, which allows us to rewrite the stationary-expectation gap as a stationary-weighted sum of row-wise expectation gaps of the Poisson solution. Applying Pinsker's inequality and Jensen's inequality then yields the stated bound. A complete proof is provided in Section D.

**A computationally tractable surrogate statistic.** Proposition 4.3 motivates a natural, computationally tractable surrogate for $L_t$ obtained by lower bounding the stationary distribution-weighted KL divergence (Definition 3.2). Concretely, we replace the unknown kernel $Q$ in (5) by its empirical estimate $\widehat{Q}_t$, maximize the resulting bound over the choice of test function $g$ for each fixed $P \in \mathcal{P}$, and then take the infimum over $P \in \mathcal{P}$. This leads to the statistic

$$\underline{L}_t := \inf_{P \in \mathcal{P}} \sup_{g:[m] \to \mathbb{R}} \frac{\left( \mathbb{E}_{\pi_{\widehat{Q}_t}}[g] - \mathbb{E}_{\pi_P}[g] \right)^2}{2\|\omega_{P,g}\|_\infty^2} \quad (6)$$

$$= \inf_{P \in \mathcal{P}} \frac{\left( \mathbb{E}_{\pi_{\widehat{Q}_t}}[g^*] \right)^2}{2\|\omega^*(\mu^*)\|_\infty^2} \quad (7)$$

where $\pi_{\widehat{Q}_t}$ denotes a stationary distribution of $\widehat{Q}_t$ (whenever it exists), $\omega_{P,g}$ denotes the solution to the PE for $(P, g)$ as

in (2) and $g^* = (I - P)\omega^*$. The vector $\omega^*(\mu^*) \in [-1, 1]^m$ is defined coordinate-wise for a threshold $\mu^*$ as:

$$\omega_i^*(\mu^*) = \begin{cases} +1, & a_i > \mu^* \pi_P(i) \\ -1, & a_i < \mu^* \pi_P(i) \\ t_i, & a_i = \mu^* \pi_P(i) \end{cases}$$

where $\mu^* \in \mathbb{R}$ is chosen such that the probability masses strictly above and strictly below the threshold are bounded by $1/2$:

$$\sum_{i:a_i > \mu^* \pi_P(i)} \pi_P(i) \leq \frac{1}{2} \quad \text{and} \quad \sum_{i:a_i < \mu^* \pi_P(i)} \pi_P(i) \leq \frac{1}{2},$$

and $t_i \in [-1, 1]$ are chosen to satisfy the constraint $\langle \pi_P, \omega^* \rangle = 0$. Equivalently, for each candidate null model $P$, $\underline{L}_t$ selects the function $g^*$ that maximizes the normalized squared discrepancy between stationary expectations under $\widehat{Q}_t$ and $P$, and then reports the least favorable value over $P \in \mathcal{P}$. The inner supremum in (6) can be solved in closed form to give (7) (see, Proposition D.1). Note that $g^*$ implicitly depends on $P$. The proof of the simplification of the inner supremum problem proceeds via reparameterizing the optimization variable using the Poisson equation, which transforms the problem into a constrained linear program and solving the dual problem. We refer the reader to Appendix D for a detailed proof.

Thresholding $\underline{L}_t$ therefore yields a computationally efficient alternative to thresholding $L_t$. Since $\underline{L}_t$ is constructed via a lower bound on the Markov divergence, the resulting test is conservative. In particular, thresholding $\underline{L}_t$ preserves $\alpha$-correctness, but may increase the expected stopping time; the magnitude of this increase depends on the tightness of the lower bound. We note that the i.i.d. literature suggests that surrogate statistics based on Pinsker-type relaxations can still lead to order-optimal procedures, albeit with a sub-optimal constant factor. Classical examples include the regret guarantees of UCB (Auer et al., 2002) vs KL-UCB (Cappé et al., 2013; Agrawal et al., 2021a), and IMED (Honda & Takemura, 2010); see also Agrawal et al. (2021b, Appendix J). A similar phenomenon may hold in the Markovian setting as well, however, a precise characterization of the resulting sample-complexity gap is beyond the scope of this work.

In addition to enabling a computationally tractable test, Proposition 4.3 also improves *interpretability*. While Theorem 4.1 characterizes the asymptotic sample complexity through $\inf_{P \in \mathcal{P}} D_{\mathcal{M}}(Q, P)$, this quantity can be abstract in applications. In Section 5.1, we give an example where Proposition 4.3 yields a natural lower bound in terms of a squared sub-optimality gap.

## 4.2. Extensions to Two-Sided Testing

In this section, we show how the one-sided (power-one, $\alpha$-correct) sequential tests studied throughout this work can be combined to construct a two-sided sequential test. We begin by formalizing two-sided testing between $\mathcal{P}$ (null) and $\mathcal{Q}$ (alternative).

For $\alpha, \beta > 0$, a level-$(\alpha, \beta)$ sequential test consists of a stopping time $\tau_{\alpha,\beta}$ and a decision rule $D \in \{0, 1\}$, where $D = 0$ and $D = 1$ correspond to selecting the null and the alternative, respectively. Recall that a one-sided test is specified solely by a stopping time $\tau_\alpha$, with rejection of the null upon stopping. For any initial distribution $\mu \in \Delta_m$, a two-sided test $(D, \tau_{\alpha,\beta})$ satisfies

$$\mathbb{P}_{P,\mu}[D(\tau_{\alpha,\beta}) = 1] \leq \alpha \qquad \forall\, P \in \mathcal{P},$$
$$\mathbb{P}_{Q,\mu}[D(\tau_{\alpha,\beta}) = 0] \leq \beta \qquad \forall\, Q \in \mathcal{Q}.$$

In the following theorem, we present a lower bound and a test that achieves it asymptotically as $(\alpha, \beta) \to 0$.

**Theorem 4.4.** *For any ergodic $Q \in \mathcal{Q}$, $P \in \mathcal{P}$, and $\alpha, \beta \in (0, 0.5)$, any level-$(\alpha, \beta)$ two-sided test with stopping time $\tau_{\alpha,\beta}$ such that $\mathbb{E}[\tau_{\alpha,\beta}] < \infty$ under both the null and the alternative satisfies*

$$\mathbb{E}_Q[\tau_{\alpha,\beta}] \geq \left( \frac{d(\beta, 1-\alpha)}{D_{\mathcal{M}}^{\inf}(Q, \mathcal{P})} - \frac{2C_Q}{\pi_Q^*} \right)^+,$$

$$\mathbb{E}_P[\tau_{\alpha,\beta}] \geq \left( \frac{d(\alpha, 1-\beta)}{D_{\mathcal{M}}^{\inf}(P, \mathcal{Q})} - \frac{2C_P}{\pi_P^*} \right)^+,$$

*where $\pi_Q^* = \min_{i \in [m]} \pi_Q(i)$, $\pi_P^* = \min_{i \in [m]} \pi_P(i)$, and $C_P, C_Q$ are as defined in Proposition 3.1.*

*Furthermore, there exists a level-$(\alpha, \beta)$ two-sided test for compact $\mathcal{P}, \mathcal{Q} \subset \mathcal{M}$ which is asymptotically optimal:*

$$\lim_{\alpha,\beta \to 0} \frac{\mathbb{E}_Q[\tau_{\alpha,\beta}]}{\log(1/\alpha)} = \frac{1}{D_{\mathcal{M}}^{\inf}(Q, \mathcal{P})} \qquad \forall\ \text{ergodic } Q \in \mathcal{Q},$$

$$\lim_{\alpha,\beta \to 0} \frac{\mathbb{E}_P[\tau_{\alpha,\beta}]}{\log(1/\beta)} = \frac{1}{D_{\mathcal{M}}^{\inf}(P, \mathcal{Q})} \qquad \forall\ \text{ergodic } P \in \mathcal{P}.$$

We establish the non-asymptotic lower bound on the expected stopping time using arguments analogous to those employed for one-sided tests. A two-sided sequential test can be constructed via a simple and intuitive approach: running two one-sided tests in parallel—one testing $\mathcal{P}$ against $\mathcal{Q}$ and the other testing $\mathcal{Q}$ against $\mathcal{P}$. This idea has previously been explored in the context of multiple hypothesis testing for i.i.d. data (Lorden, 1976; 1977; Baum & Veeravalli, 2002). We show that this technique yields an asymptotically optimal test for Markovian data.

We refer the reader to Section C for a detailed proof of Theorem 4.4.

## 5. Applications

We now present two concrete applications of our framework: (i) testing misspecification in MCMC samplers, and (ii) testing linear transition dynamics in MDPs.

### 5.1. Testing Misspecification in MCMC Samplers

Let $\pi^\top \in \Delta_m$ be a known, strictly positive target distribution. In MCMC, one constructs an ergodic Markov chain with transition matrix $P$ satisfying the stationarity condition $\pi P = \pi$. Expectations under $\pi$ are then estimated via time averages of a function $f$ along the trajectory.

In practice—especially with approximate kernels or black-box samplers—it is often necessary to validate whether the observed path $X_1, X_2, \ldots$ could plausibly have been generated by a Markov chain whose stationary distribution is $\pi^\top$. If the underlying kernel violates $\pi P = \pi$, then the resulting estimates are asymptotically biased. Consequently, it is important to detect misspecification quickly whenever it occurs.

Related work on testing MCMC procedures typically studies threshold tests for stationary expectations of a fixed function (Gyori & Paulin, 2015; Rabinovich et al., 2020). Our goal is different: we test misspecification of the *transition structure* without committing to a specific test function. We also emphasize that our objective is not to test convergence diagnostics for MCMC (Roy, 2020), but rather to test whether the observed samples could converge to the desired stationary distribution.

**Problem formulation.** Define the set of valid transition matrices
$$\mathcal{P}_\pi \triangleq \{P \in \mathcal{M} : \ \pi P = \pi\}.$$
We test $\mathcal{P} = \mathcal{P}_\pi$ against $\mathcal{Q} = \mathcal{P}_\pi^{\complement}$.

The one-sided nature of our framework is well-suited to this setting. Under the null, continued sampling is beneficial since it reduces the variance of the MCMC estimator. Our test guarantees that, when there is no misspecification, the chain is not stopped with probability at least $1 - \alpha$, allowing estimation to proceed in parallel with testing. Under misspecification, further sampling only increases the computational cost of producing biased samples. In this case, our test stops with probability 1, preventing continued waste of computation.

To instantiate the test, we require compactness of $\mathcal{P}_\pi$, which we establish in Lemma E.1 in Section E. This yields the following corollary to Theorem 4.1.

**Corollary 5.1.** *For any ergodic $Q \in \mathcal{P}_\pi^{\complement}$ with stationary distribution $\pi_Q \neq \pi$, the stopping time $\tau_\alpha$ satisfies*

$$\limsup_{\alpha \to 0} \frac{\mathbb{E}_Q[\tau_\alpha]}{\log(1/\alpha)} \leq \frac{1}{D_{\mathcal{M}}^{\inf}(Q, \mathcal{P}_\pi)}.$$

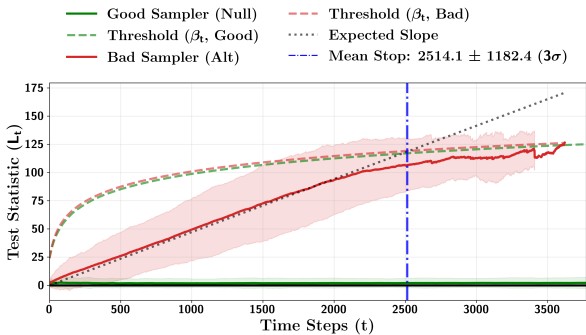

Figure 1. Test statistic trajectories over time for $Q_{\text{bad}}$ (red) and $Q_{\text{good}}$ (green), aggregated over 100 runs. Solid curves and shaded regions show the mean $\pm 3\sigma$; dotted colored curves denote the boundary. The dotted black line shows the theoretical slope $D_{\mathcal{M}}^{\inf}(Q_{\text{bad}}, \mathcal{P}_\pi)$, and the vertical line marks the mean stopping time for $Q_{\text{bad}}$.

**Interpretation in terms of estimation bias.** As discussed in Section 4.1, Proposition 4.3 can be used to obtain interpretable lower bounds on $D_{\mathcal{M}}(Q, P)$. In this application, for any function of interest $g$, the quantity

$$\Delta \triangleq \left| \mathbb{E}_{\pi_Q}[g] - \mathbb{E}_\pi[g] \right|$$

is precisely the asymptotic bias of the MCMC estimator under $Q$. Applying Proposition 4.3 yields that $D_{\mathcal{M}}(Q, P)$ can be lower bounded by a term of order $\Delta^2$, and hence the dominant scaling of the complexity term behaves as $1/D_{\mathcal{M}}(Q, P) = O(1/\Delta^2)$.

**Experimental validation.** We fix a target distribution $\pi \in \Delta_5$ and set $\alpha = 0.05$. We compare two kernels: a valid sampler $Q_{\text{good}} \in \mathcal{P}_\pi$ and a misspecified sampler $Q_{\text{bad}} \notin \mathcal{P}_\pi$. As shown in Figure 1, under the alternative ($Q_{\text{bad}}$) the statistic grows approximately linearly and tracks the rate $D_{\mathcal{M}}^{\inf}(Q_{\text{bad}}, \mathcal{P}_\pi)$. Under the null ($Q_{\text{good}}$), the statistic remains below the boundary. Across all trials, we observed no false rejections, suggesting that the boundary may be conservative in practice. Additional details are provided in Section G.

## 5.2. Testing Linearity of MDPs

We consider the problem of testing a commonly used structural assumption in reinforcement learning: that the transition dynamics and rewards admit a linear parameterization with respect to a known feature map.

Formally, consider an infinite-horizon MDP M with finite state space $\mathcal{S}$ and finite action space $\mathcal{A}$. The MDP is specified by transition probabilities $p_{\mathsf{M}}(s, a, s')$ and reward distributions $q_{\mathsf{M}}(s, a)$, with expected reward $r_{\mathsf{M}}(s, a) = \mathbb{E}[q_{\mathsf{M}}(s, a)]$. A policy $\pi_{\mathsf{M}}(a'|s')$ specifies the probability of selecting action $a'$ in state $s'$.

Let $T_{\mathsf{M}} \in \mathbb{R}^{|\mathcal{S}||\mathcal{A}| \times |\mathcal{S}|}$ denote the transition matrix with en-

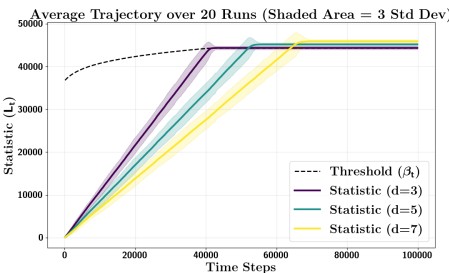

Figure 2. Mean statistic trajectory aggregated over 20 runs. Shaded regions indicate $\pm 3$ standard deviations.

tries $T_{\mathsf{M}}((s, a), s') \triangleq p_{\mathsf{M}}(s, a, s')$. Let $\Pi_{\mathsf{M}} \in \mathbb{R}^{|\mathcal{S}| \times |\mathcal{S}||\mathcal{A}|}$ denote the policy matrix with entries $\Pi_{\mathsf{M}}(s, (s', a')) \triangleq \pi_{\mathsf{M}}(a'|s) \mathbb{I}\{s = s'\}$.

**Definition 5.2** (Linear MDP (Jin et al., 2020)). An MDP M is *linear* if there exist $\mu_{\mathsf{M}} \in \mathbb{R}^{|\mathcal{S}| \times d}$ and $\theta_{\mathsf{M}} \in \mathbb{R}^d$ such that

$$T_{\mathsf{M}} = \Phi \mu_{\mathsf{M}}^\top, \qquad r_{\mathsf{M}} = \Phi \theta_{\mathsf{M}},$$

where $\Phi \in \mathbb{R}^{|\mathcal{S}||\mathcal{A}| \times d}$ is a known feature matrix whose row for $(s, a)$ equals $\phi(s, a)^\top$ for a given map $\phi : \mathcal{S} \times \mathcal{A} \to \mathbb{R}^d$. We assume $\|\theta_{\mathsf{M}}\|_2 \le \sqrt{d}$, $\|\phi(s, a)\|_2 \le 1$ for all $(s, a)$, and $\|\mu_{\mathsf{M}}(s)\|_2 \le \sqrt{d}$ for all $s \in \mathcal{S}$.

Executing policy $\pi_{\mathsf{M}}$ in M induces a Markov chain $P_{\mathsf{M}, \pi}$ on state–action pairs with transition matrix

$$P_{\mathsf{M}, \pi}\big((s, a), (s', a')\big) = p_{\mathsf{M}}(s, a, s')\, \pi_{\mathsf{M}}(a'|s').$$

We assume the induced chain is ergodic. For a linear MDP, this induced kernel can be written as

$$P_{\mathsf{M}, \pi} = \Phi \mu_{\mathsf{M}}^\top \Pi_{\mathsf{M}}.$$

Since $\Phi$ and $\Pi_{\mathsf{M}}$ are known, linearity imposes explicit structural constraints on the induced transitions. We encode these constraints through the null class

$$\mathcal{P}_L = \left\{ P : \ P = \Phi \mu^\top \Pi_{\mathsf{M}}, \ \mu \in \mathbb{R}^{|\mathcal{S}| \times d}, \ \|\mu\|_{2,\infty} \le \sqrt{d} \right\}.$$

We test $\mathcal{P} = \mathcal{P}_L$ against $\mathcal{Q} = \mathcal{P}_L^{\complement}$. We establish compactness of $\mathcal{P}_L$ in Lemma E.2 in Section E, which yields the following corollary of Theorem 4.1.

**Corollary 5.3.** *Suppose M is not linear (with respect to $\Phi$), and that policy $\pi_{\mathsf{M}}$ induces an ergodic Markov chain $Q_{\mathsf{M}, \pi}$ on state–action pairs. Then*

$$\limsup_{\alpha \to 0} \frac{\mathbb{E}_{Q_{\mathsf{M}, \pi}}[\tau_\alpha]}{\log(1/\alpha)} \le \frac{1}{D_{\mathcal{M}}^{\inf}(Q_{\mathsf{M}, \pi}, \mathcal{P}_L)}.$$

Here, $D_{\mathcal{M}}^{\inf}(Q_{\mathsf{M}, \pi}, \mathcal{P}_L)$ quantifies how far the induced dynamics are from admitting the prescribed linear factorization.

**Experimental validation.** We evaluate our procedure on the Mountain Car environment as implemented in `gym` (Towers et al., 2024). We discretize the state space ($|\mathcal{S}| = 8 \times 8$) and use $|\mathcal{A}| = 3$ actions with a uniformly random policy. We generate radial basis function features of varying dimensions and plot the statistic over time in Figure 2. In all cases, the statistic grows steadily, indicating misspecification and rejection of the linear MDP hypothesis. The lower-dimensional representation ($d = 3$) exhibits the steepest growth and thus rejects earlier than higher-dimensional representations ($d = 5, 7$). Additional experimental results, including a comparison with the algorithm of Fields et al. (2025), are in Section G.

## 6. Discussion and Future Works

We developed a rigorous framework for one-sided sequential hypothesis testing with Markovian data. Our results substantially generalized prior work by allowing a composite null to be tested against a disjoint, composite alternative. We introduced new tools that yielded the first non-asymptotic, instance-dependent lower bounds in this setting, and we proposed a martingale-based test whose asymptotic sample complexity matched these bounds. We illustrated the framework through applications to misspecification testing in MCMC and to structural testing in reinforcement learning. We also established a Pinsker-type inequality for Markov divergences via the Poisson equation, which may be of independent interest.

Several directions remain open. First, it would be valuable to extend the techniques of Fields et al. (2025) to the general composite setting; this likely requires new low-regret learning procedures under Markovian dependence. Second, our results highlight a statistical–computational tradeoff: while the approximation in Section 4.1 yields computationally tractable tests, it may be sample-inefficient. It remains unclear whether one can achieve both computational efficiency and statistical optimality for general (possibly non-convex) $\mathcal{P}$. Along the same lines, it would be interesting to understand the impact of using the mixture martingale $M_t$ (see Section B.3) directly instead of the computationally tractable lower bound. Third, it is natural to move beyond finite state spaces to countable-state Markov chains. Finally, an important extension is sequential testing under hidden Markov observations, where only a noisy function of the latent chain is observed and the resulting process need not be Markov. This raises major analytical challenges for both $\alpha$-correct martingale constructions and sharp, instance-dependent lower bounds.

## Impact Statement

This work advances the theoretical foundations of machine learning. While hypothesis testing, MCMC, and reinforcement learning underpin many societal applications (e.g., autonomous driving and healthcare), our contribution is methodological and theoretical in nature. Accordingly, we expect the direct ethical impacts of this paper to be limited, though the results may indirectly inform the design of provably efficient algorithms for such applications.

## Acknowledgements

The authors would like to thank the anonymous reviewers for their insightful comments and suggestions. SA acknowledges the generous support from the Pratiksha Trust, Bangalore, through the Young Investigator Award, and the DST INSPIRE Faculty Grant IFA24-ENG-389. DB acknowledges the support of the ANR JCJC project REPUBLIC (ANR-22-CE23-0003-01) and PEPR project FOUNDRY (ANR23-PEIA-0003). We also acknowledge the Inria-IISc associate team FOSSIL (DRI-012643) for supporting the collaboration.

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

# A. Proofs of Results Appearing in Section 3

This appendix builds the requisite tools and then proves the lower bound, Theorem 3.3. Section A.1 provides a recap of the results from Moustakides (1999) on a Wald-type identity for Markovian data, and an application of the same to our problem setting. We show that the latter leads to certain auxiliary terms which arise as solutions to the Poisson equation (PE). In Section A.2, we build the primary tool required to obtain a bound on the magnitude of the auxiliary terms via the spectral properties of the underlying Markov chain. Finally, we prove the lower bound in Section A.3.

## A.1. Wald's Lemma for Markovian Data

Towards building our machinery, we first recap the Wald's lemma for Markov chains from Moustakides (1999).

**Lemma A.1** (Moustakides (1999, Lemma 2.3)). *Let* $P \in \mathcal{M}$ *be aperiodic and irreducible, with a unique stationary distribution* $\pi \in \mathbb{R}^{1 \times m}$. *Let* $f \in \mathbb{R}^m$. *Consider the PE with respect to the unknown* $\nu \in \mathbb{R}^m$, *given by*

$$(P - I)\nu = -(P - \mathbf{1}_{m \times 1}\pi)f,$$

*where* $\mathbf{1}$ *denotes the all-ones vector, and* $\nu$ *satisfies the constraint* $\pi \nu = 0$. *Then, the above system of equations possesses a solution given by*

$$\nu = \sum_{n=1}^{\infty} (P^n - \mathbf{1}\pi)f.$$

We note here that the form of the PE appearing in Lemma A.1 differs slightly from the standard form in that if $\omega = \omega^*$ denotes a solution to the (standard form of) PE

$$(I - P)\omega = f - (\pi f)\mathbf{1},$$

then $\nu$ and $\omega^*$ satisfy the relation $\nu = P\omega^*$. Accordingly, the statement of Wald's lemma in Moustakides (1999) can be modified to use the solution $\omega^*$ of the standard form of the PE with minor modifications to the martingale construction therein. We record this in the following result and provide a proof of the same for completeness.

**Theorem A.2** (Wald's Lemma for Markov Chains). *Let* $\{X_n\}_{n \geq 0}$ *be an irreducible, aperiodic Markov chain on a finite state space with* $m$ *states and transition matrix* $P$. *Let* $\pi$ *denote the unique stationary distribution of* $P$. *Let* $f \in \mathbb{R}^m$ *be any cost function, and let* $N$ *be a stopping time with respect to the natural filtration generated by the process* $\{X_n\}_{n \geq 0}$, *with* $\mathbb{E}[N] < +\infty$. *Define the total accumulated cost* $S_N := \sum_{k=0}^{N-1} f(X_k)$. *Let* $\omega_{P,f} \in \mathbb{R}^m$ *be a solution to the PE* $(I - P)\omega_{P,f} = f - (\pi f)\mathbf{1}$. *Then, we have*

$$\mathbb{E}[S_N] = (\pi f)\,\mathbb{E}[N] + \mathbb{E}[\omega_{P,f}(X_0) - \omega_{P,f}(X_N)].$$

*Proof.* For the rest of the proof, we drop the subscript in $\omega_{P,f}$. Define the martingale process $\{U_n\}_{n \geq 0}$ as

$$U_0 = \omega(X_0), \qquad U_n = \omega(X_n) + \sum_{k=0}^{n-1} (f(X_k) - \pi f), \qquad n \geq 1.$$

To verify the martingale property, defining $\mathcal{F}_n = \sigma(X_0, X_1, \ldots, X_n)$ for each $n \geq 0$, we have

$$\mathbb{E}[U_{n+1} \mid \mathcal{F}_n] = \mathbb{E}\left[\omega(X_{n+1}) + \sum_{k=0}^{n} (f(X_k) - \pi f) \,\bigg|\, \mathcal{F}_n\right],$$

which may be rewritten as

$$\mathbb{E}[U_{n+1} \mid \mathcal{F}_n] = \mathbb{E}[\omega(X_{n+1}) \mid X_n] + (f(X_n) - \pi f) + \sum_{k=0}^{n-1} (f(X_k) - \pi f).$$

Observe that $\mathbb{E}[\omega(X_{n+1}) \mid X_n] = (P\omega)(X_n)$. Thus,

$$\mathbb{E}[U_{n+1} \mid \mathcal{F}_n] = (P\omega)(X_n) + f(X_n) - \pi f + \sum_{k=0}^{n-1} (f(X_k) - \pi f).$$

Rearranging terms in the PE $(I - P)\omega = f - (\pi f)\mathbf{1}$, we get

$$(P\omega)(X_n) = \omega(X_n) - (f(X_n) - \pi f),$$

from which it follows that

$$\mathbb{E}[U_{n+1} \mid \mathcal{F}_n] = [\omega(X_n) - (f(X_n) - \pi f)] + (f(X_n) - \pi f) + \sum_{k=0}^{n-1}(f(X_k) - \pi f)$$

$$= \omega(X_n) + \sum_{k=0}^{n-1}(f(X_k) - \pi f)$$

$$= U_n,$$

thereby proving that $\{U_n\}_{n \geq 0}$ is a martingale.

Noting that the state space is finite, we have that $\omega, f$ are bounded. Then, there exists $K < +\infty$ such that for all $n \geq 1$,

$$|U_{n+1} - U_n| = |\omega(X_{n+1}) - \omega(X_n) + f(X_n) - \pi f| \leq K \qquad \text{a.s.,}$$

thus proving that $\{U_n\}_{n \geq 0}$ has bounded increments. Given that $N$ is a stopping time with $\mathbb{E}[N] < \infty$ and the martingale $\{U_n\}_{n \geq 0}$ has bounded increments, we apply the optional stopping theorem to deduce that

$$\mathbb{E}[U_N] = \mathbb{E}[U_0],$$

which in turn is equivalent to

$$\mathbb{E}[\omega(X_N)] + \mathbb{E}\left[\sum_{k=0}^{N-1} f(X_k)\right] - (\pi f)\,\mathbb{E}[N] = \mathbb{E}[\omega(X_0)].$$

Rearranging terms, we arrive at the desired result. $\qquad \square$

## A.2. Bounding the Magnitude of a Solution to the Poisson Equation via Spectral Properties

To apply Theorem A.2 later on, we desire control over the magnitude of $\omega^* = \omega_{P,f}$, the solution to the PE for the ergodic Markov chain with transition matrix $P$ and the function $f$. In the following result below, we derive a bound on the magnitude of $\omega_{P,f}$ (as given in (2)) that depends on $P$ and $f$.

As a first step, we record the following corollary to (Paulin, 2015, Proposition 3.4).

**Corollary A.3.** *Let $P$ be an ergodic transition matrix on a finite state space $[m]$ with unique stationary distribution $\pi$. Let $\gamma_{\mathrm{ps}}$ be the pseudo-spectral gap. Then, for any $n \geq 1$ and $x \in [m]$, we have*

$$\|P^n(x, \cdot) - \pi\|_{\mathrm{TV}} \leq \frac{1}{2(1 - \gamma_{\mathrm{ps}})^{1/(2\gamma_{\mathrm{ps}})}\sqrt{\pi(x)}}(1 - \gamma_{\mathrm{ps}})^{n/2}. \tag{8}$$

*Remark* A.4. It is worthwhile to compare (8) with (Fill, 1991, Theorem 2.1) which gives a similar bound but in terms of the second largest eigenvalue of the multiplicative reversibilization of $P$ (i.e., $P^*P$). However, as noted in (Paulin, 2015, Remark 3.2), there exist cases where the second largest eigenvalue is zero, while the pseudo-spectral gap is strictly positive.

*Proof of Corollary A.3.* Observe that a finite state, ergodic Markov chain is uniformly ergodic (Douc et al., 2018, Exercise 15.6), and consider the initial distribution $q$ of the Markov chain localized at $x$, i.e., $q(x) = 1$ and $q(y) = 0$ for all $y \neq x$. Applying (Paulin, 2015, Proposition 3.4), we have

$$\begin{aligned}
\|P^n(x, \cdot) - \pi\|_{\mathrm{TV}} &\leq \frac{1}{2}(1 - \gamma_{\mathrm{ps}})^{(n-1/\gamma_{\mathrm{ps}})/2}\sqrt{\frac{1}{\pi(x)} - 1} \\
&\leq \frac{1}{2(1 - \gamma_{\mathrm{ps}})^{1/(2\gamma_{\mathrm{ps}})}\sqrt{\pi(x)}}(1 - \gamma_{\mathrm{ps}})^{n/2},
\end{aligned}$$

where the second line follows by noting that $\sqrt{1 - \pi(x)} \leq 1$. $\qquad \square$

With the above corollary in hand, we provide the proof of Proposition 3.1 below.

*Proof of Proposition 3.1.* We use the form of $\omega = \omega_{P,f}$ given in (2), i.e.

$$\omega = \sum_{n=0}^{\infty}(P^n - \mathbf{1}\pi)f.$$

More explicitly, for each $x \in [m]$, we have

$$\omega(x) = \sum_{n=0}^{\infty}\sum_{y\in[m]}(P^n(x,y) - \pi(y))\,f(y).$$

Taking absolute values on both sides and applying the triangle inequality, we get

$$|\omega(x)| \leq \sum_{n=0}^{\infty}\sum_{y\in[m]}|P^n(x,y) - \pi(y)|\,|f(y)|$$

$$\leq 2\|f\|_{\infty}\sum_{n=0}^{\infty}\|P^n(x,\cdot) - \pi\|_{\mathrm{TV}}.$$

Assuming that $\gamma_{\mathrm{ps}} \in (0,1)$ and using Corollary A.3 to bound the total variation terms, we have

$$|\omega(x)| \leq \frac{\|f\|_{\infty}}{(1-\gamma_{\mathrm{ps}})^{1/(2\gamma_{\mathrm{ps}})}\sqrt{\pi(x)}}\sum_{n=0}^{\infty}(1-\gamma_{\mathrm{ps}})^{n/2}$$

$$= \frac{\|f\|_{\infty}}{(1-\gamma_{\mathrm{ps}})^{1/(2\gamma_{\mathrm{ps}})}\sqrt{\pi(x)}}\left(\frac{1}{1-\sqrt{1-\gamma_{\mathrm{ps}}}}\right).$$

Replacing $\pi(x)$ with the $\pi_* = \min_{x\in[m]}\pi(x)$ gives an upper bound that is valid for any $x \in [m]$, thereby giving us the desired result.

We now show boundedness of $\omega = \omega_{P,f}$ for the corner cases when $\gamma_{\mathrm{ps}} \in \{0,1\}$. Recall that

$$\gamma_{\mathrm{ps}}(P) := \max_{k\geq 1}\frac{\gamma((P^*)^k P^k)}{k},$$

From (Fill, 1991, Section 2.1), we have that $(P^*)^k P^k$ is reversible for each $k \geq 1$ and has all eigenvalues in $[0,1]$, thus implying that $\gamma_{\mathrm{ps}} \in [0,1]$.

- $\gamma_{\mathrm{ps}} = 0$: We argue that this case is not possible for ergodic, finite state Markov chains. From the convergence theorem for ergodic Markov chains (Levin et al., 2017, Theorem 4.9), we know that the mixing time, $t_{\mathrm{mix}}$, of an ergodic matrix is finite. Paulin (2015, Proposition 3.4) shows that $\gamma_{\mathrm{ps}} \geq 1/(2t_{\mathrm{mix}})$ for ergodic Markov chains. Thus, we must have that $\gamma_{\mathrm{ps}} > 0$.

- $\gamma_{\mathrm{ps}} = 1$: We show below that this case corresponds to the i.i.d. setting, and that in this case, $\omega_{P,f} = f - (\pi f)\mathbf{1}$.

   Observe that if $\gamma_{\mathrm{ps}} = \max_{k\geq 1}\frac{\gamma((P^*)^k P^k)}{k} = 1$, we must have that $\gamma_{\mathrm{ps}} = \gamma(P^*P) = 1$, as $\frac{\gamma((P^*)^k P^k)}{k} \in [0,1/k]$ for any $k$ (noting that $\gamma((P^*)^k P^k) \in [0,1]$ for all $k \geq 1$). From the definition of spectral gap, we have $1 - \lambda_2(P^*P) = 1$, thereby implying that $\lambda_2 = \lambda_2(P^*P) = 0$. Using the fact that all eigenvalues of $P^*P$ lie in $[0,1]$, we must have $0 = \lambda_2 = \lambda_3 \cdots = \lambda_m$. Applying (Fill, 1991, Theorem 2.1), we have that $\|P^n(x,\cdot) - \pi\|_{\mathrm{TV}} \leq 0$ for all $n \geq 1$ and $x \in [m]$, thus implying that $P(x,\cdot) = \pi$ for all $x \in [m]$. That is, $P$ has identical rows, all of which match with the stationary distribution, thus implying that the random variables of the process $\{X_n\}_{n\geq 0}$ are independent and identically distributed (i.i.d.), with distribution $\pi$. In this case, the PE gives

$$(I - P)\omega_{P,f} = f - (\pi f)\mathbf{1} \quad \implies \quad \omega_{P,f} - (\pi\omega_{P,f})\mathbf{1} = f - (\pi f)\mathbf{1}.$$

We note that the solution to PE is unique on the subspace of $\mathbb{R}^m$ orthogonal to the all ones vector, $\mathbf{1}$, and the above equation shows that any $\omega_{P,f}$ and $f$ are equal on this subspace. We can then choose an $\omega_{P,f}$ such that its projection along $\mathbf{1}$ is equal to zero. This gives us $\omega_{P,f} = f - (\pi f)\mathbf{1}$, which we note is consistent with the solution given as (2) in the main text. The solution now follows from observing: $\|f\|_\infty \geq \pi f$ and $|f(i)| \leq \|f\|_\infty \ \forall i \in [m]$. Applying the triangle inequality, we get: $|f(i) - \pi f| \leq 2\|f\|_\infty \ \forall i \in [m]$ and we obtain the result.

This completes the proof. $\qquad\square$

To summarize the above discussion, we have that for an ergodic Markov chain $P$ and any function $f : [m] \to \mathbb{R}$,

$$\frac{\|\omega_{P,f}\|_\infty}{\|f\|_\infty} \leq \begin{cases} \dfrac{1}{(1-\gamma_{\mathrm{ps}})^{1/(2\gamma_{\mathrm{ps}})}\sqrt{\pi_*}} \left( \dfrac{1}{1-\sqrt{1-\gamma_{\mathrm{ps}}}} \right), & \gamma_{\mathrm{ps}} \in (0,1) \\ 2, & \gamma_{\mathrm{ps}} = 1. \end{cases} \tag{9}$$

*Remark* A.5 (Choice of Spectral Gap). One may also bound the solution to the PE with a different notion of the spectral gap, say $\gamma_c$, defined by Chatterjee (2025). In particular, we have (Chatterjee, 2025, Equation 3.1)

$$\|\omega\|_{2,\pi} \leq \frac{1}{\gamma_c}\|f - (\pi f)\mathbf{1}\|_{2,\pi},$$

where $\|a\|_{2,\pi} := \sum_{x \in [m]} \pi(x)(a(x))^2$. However, the key difference arises in the fact that this bound is stated in terms of the norm of the centered version of the function, i.e., $\|f - (\pi f)\mathbf{1}\|_{2,\pi}$. Our bound, on the contrary, is in terms of the norm of the uncentered function, i.e., $\|f\|_\infty$. It is not immediately obvious how the bound with the centered version may be used to derive a bound with the uncentered version, or vice-versa. In general, we do not know how such a result would be derived for other notions of spectral gaps.

Looking forward, we note that the analysis of the upper bound could potentially be done by a different choice of spectral gap. This can be done by substituting the relevant concentration inequalities in Section B.2 (for example, by using Huang & Li (2024, Theorem 2.4)).

### A.3. Proofs Related to the Lower Bound

We are now equipped with all the tools needed to the lower bound. We proceed as follows: first, we use Theorem A.2 to control the expected log-likelihood ratio between two data generating chains as in Fields et al. (2025, Appendix). Following this, we proceed to show the lower bound for testing $Q$ against a singleton $P \in \mathcal{P}$ via a change-of-measure argument. While the analogous proof for the independent setting (Agrawal & Ramdas, 2025, Theorem C.1) effectively stops here by taking a supremum over all elements of the null hypothesis set, we require the bounds developed in Proposition 3.1 to bound the auxiliary PE terms arising from Theorem A.2, which depend implicitly on $P$, so that we can safely maximize the lower bound over the whole of the null hypothesis set $\mathcal{P}$.

**Corollary A.6** (Expected log-likelihood ratio). *Let $\{X_n\}_{n \geq 0}$ be an ergodic Markov chain with transition matrix $Q$ on a finite state space, say $[m]$. Let $P$ be any transition matrix on the state space $[m]$, with $Q \ll P$. Let $\tau_\alpha$ be a stopping time with respect to the natural filtration generated by $\{X_n\}_{n \geq 0}$, with $\mathbb{E}[\tau_\alpha] < +\infty$. Then,*

$$\mathbb{E}_Q \left[ \sum_{i=1}^{\tau_\alpha} \log\left( \frac{Q(X_{i-1}, X_i)}{P(X_{i-1}, X_i)} \right) \right] = D_{\mathcal{M}}(Q, P)\,\mathbb{E}[\tau_\alpha] + C_{\tau_\alpha},$$

*where $C_{\tau_\alpha} := \mathbb{E}[\omega_{Q,f_P}(X_0) - \omega_{Q,f_P}(X_{\tau_\alpha})]$, $\omega_{Q,f_P}$ is a solution of the PE for the function $f_P$ where $f_P(i) = D_{\mathrm{KL}}(Q(i,\cdot), P(i,\cdot))$ for all $i \in [m]$.*

*Proof.* Consider the function:

$$f(X_n) := D_{\mathrm{KL}}(Q(X_n,\cdot), P(X_n,\cdot)) = \mathbb{E}_{X_{n+1} \sim Q(X_n,\cdot)}\left[ \log\left( \frac{Q(X_n, X_{n+1})}{P(X_n, X_{n+1})} \right) \right].$$

Using the fact that $Q \ll P$, we have $f(X_n) < +\infty$ for all $n$. Applying Theorem A.2, the desired result follows by observing that $\pi f = D_{\mathcal{M}}(Q, P)$. $\qquad\square$

We now proceed to prove Theorem 3.3.

*Proof of Theorem 3.3.* Let $\{X_n\}_{n \geq 0}$ be an ergodic Markov chain with transition matrix some $Q \in \mathcal{Q}$. If $Q \not\ll P$ for any $P \in \mathcal{P}$, then $D_{\mathcal{M}}^{\inf}(Q, \mathcal{P}) = +\infty$ and the lower bound in (3) holds trivially. Therefore, fixing $P \in \mathcal{P}$ such that $Q \ll P$, we have

$$\mathbb{E}_Q \left[ \sum_{t=1}^{\tau_\alpha} \log \left( \frac{Q(X_{t-1}, X_t)}{P(X_{t-1}, X_t)} \right) \right] = \mathrm{D_{KL}} \left( Q^{\tau_\alpha}, P^{\tau_\alpha} \right),$$

where the divergence is between the joint distributions on the $\tau_\alpha$ samples induced by the Markov chains $Q$ and $P$ respectively. From Corollary A.6, we have

$$\mathbb{E}_Q \left[ \sum_{t=1}^{\tau_\alpha} \log \left( \frac{Q(X_{t-1}, X_t)}{P(X_{t-1}, X_t)} \right) \right] = \mathbb{E}_Q \left[ \tau_\alpha \right] D_{\mathcal{M}}(Q, P) + C_{\tau_\alpha, f_P},$$

where $C_{\tau_\alpha, f_P} = \mathbb{E}[\omega_{Q, f_P}(X_0) - \omega_{Q, f_P}(X_{\tau_\alpha})]$. Applying the data processing inequality, we have

$$\mathrm{D_{KL}} \left( Q^{\tau_\alpha}, P^{\tau_\alpha} \right) \geq d(Q^{\tau_\alpha}(\mathcal{E}), P^{\tau_\alpha}(\mathcal{E})) \qquad \forall \, \mathcal{E} \in \mathcal{F}_{\tau_\alpha},$$

where $\mathcal{F}_{\tau_\alpha} := \left\{ \mathcal{E} : \mathcal{E} \cap \{\tau_\alpha = n\} \in \mathcal{F}_n \ \forall \, n \right\}$. Choosing the event $\mathcal{E} = \{\tau_\alpha < +\infty\}$, and noting that $\mathcal{E} \in \mathcal{F}_{\tau_\alpha}$, we have

$$\mathrm{KL}(Q^{\tau_\alpha}, P^{\tau_\alpha}) \geq d(1, \alpha) = \log(1/\alpha).$$

Then, for any $P \in \mathcal{P}$, we have

$$\mathbb{E}_Q \left[ \tau_\alpha \right] \geq \frac{\log(1/\alpha)}{D_{\mathcal{M}}(Q, P)} - \frac{\mathbb{E}[\omega_{Q, f_P}(X_0) - \omega_{Q, f_P}(X_\tau)]}{D_{\mathcal{M}}(Q, P)}.$$

The above bound holds for any $P \in \mathcal{P}$, however to safely take a supremum over all such $P \in \mathcal{P}$, we get the following *worst case* bound on the latter term:

$$\frac{\mathbb{E}[\omega_{Q, f_P}(X_0) - \omega_{Q, f_P}(X_{\tau_\alpha})]}{D_{\mathcal{M}}(Q, P)} \leq \frac{2\|\omega_{Q, f_P}\|_\infty}{D_{\mathcal{M}}(Q, P)} \leq \frac{2\|\omega_{Q, f_P}\|_\infty}{\pi_* \|f_P\|_\infty} \leq \frac{2C_Q}{\pi_*},$$

where the third inequality above follows by noting that

$$D_{\mathcal{M}}(Q, P) = \sum_{i \in [M]} \pi_i \, \mathrm{D_{KL}} \left( (, Q)(i, \cdot), P(i, \cdot) \right) \geq \pi_* \sum_{i \in [M]} \underbrace{\mathrm{D_{KL}} \left( (, Q)(i, \cdot), P(i, \cdot) \right)}_{f_P(i)} \geq \pi_* \|f_P\|_\infty,$$

and the last inequality follows from the definition of $C_Q$ (see Proposition 3.1). Thus, we have the worst case bound:

$$\mathbb{E}_Q \left[ \tau_\alpha \right] \geq \frac{\log(1/\alpha)}{D_{\mathcal{M}}(Q, P)} - \frac{2C_Q}{\pi_*}.$$

Noting that the above inequality holds for any $P \in \mathcal{P}$, and replacing $D_{\mathcal{M}}(Q, P)$ with $\inf_{P \in \mathcal{P}} D_{\mathcal{M}}(Q, P)$, we arrive at the desired result. Since our worst-case bound on the PE terms may be loose in certain cases, we take a maximum between our obtained bound and 0 (since $\tau_\alpha \geq 0$) to ensure the bound isn't vacuous in instances where $\frac{\log(1/\alpha)}{D_{\mathcal{M}}^{\inf}(Q, \mathcal{P})} < \frac{2C_Q}{\pi_*}$. $\qquad \square$

### A.4. Suboptimality of Existing Lower Bounds

We note here that we could have alternatively used Al Marjani & Proutiere (2021, Lemma 1) to get a bound of the following form:

$$\mathbb{E}_Q \left[ \tau_\alpha \right] \geq \frac{\log(1/\alpha)}{\sup_{\omega \in \Delta_m} \inf_{P \in \mathcal{P}} \left( \sum_{i \in [m]} \omega_i \mathrm{D_{KL}} \left( Q(x, \cdot), P(x, \cdot) \right) \right)}. \tag{10}$$

While this is a valid lower bound, we demonstrate that it is suboptimal in our setting with the help of a simple example. Consider testing a singleton null, i.e. $\mathcal{P} = \{P\}$ against a problem instance $Q$ (with corresponding stationary distribution $\pi$).

Let $f \in \mathbb{R}^m$ be the vector corresponding to the row KL divergences, i.e. $f(i) = \mathrm{D_{KL}}\left(Q(i,\cdot), P(i,\cdot)\right)$. We choose $P$ and $Q$ such that $f$ is not a constant vector, i.e. $\exists\, i, j \in [m] : \mathrm{D_{KL}}\left(Q(i,\cdot), P(i,\cdot)\right) < \mathrm{D_{KL}}\left(Q(j,\cdot), P(j,\cdot)\right)$.

In this example, the term in the denominator of (10) becomes $\sup_{\omega \in \Delta_m} \inf_{P \in \mathcal{P}} \omega^\top f$. Since the null is a singleton set, the inner infimum disappears, and it is easy to see that $\sup_{\omega \in \Delta_m} \omega^\top f = \|f\|_\infty$. Thus, (10) becomes:

$$\mathbb{E}_Q\left[\tau_\alpha\right] \geq \frac{\log(1/\alpha)}{\|f\|_\infty}. \tag{11}$$

On the contrary, our proposed lower bound (Theorem 3.3) gives:

$$\mathbb{E}_Q\left[\tau_\alpha\right] \geq \frac{\log(1/\alpha)}{\pi f} - \frac{2C_Q}{\pi_*}. \tag{12}$$

Observe that $\pi f = \sum_{i \in [m]} \pi_i \mathrm{D_{KL}}\left(Q(i,\cdot), P(i,\cdot)\right) < \max_i \mathrm{D_{KL}}\left(Q(i,\cdot), P(i,\cdot)\right) = \|f\|_\infty$, where the strict inequality comes from the fact that $\min_i \pi_i > 0$ and the fact that $f$ is not constant. Thus, the dominant term in (12) is strictly larger than (11). Taking $\alpha \to 0$, we can see that the bound given by (11) is worse by a multiplicative factor.

To understand why this approach is loose in our setup but not in the context of best policy identification in MDPs, one needs to understand a subtle difference in setups.

Suppose the supremum in (10) is achieved by $\omega^* \in \Delta_m$. In the context of best policy identification in MDPs, one interprets $\omega^*$ as the optimal proportion in which one should visit states and the design of the optimal algorithm focuses on tracking this proportion. On the other hand, in our setup (as well as that in Moulos (2019); Karthik et al. (2024); Ariu et al. (2025)), we do not decide the proportion in which to visit the states of the Markov chain, it is, in fact, specified by the problem instance! Owing to this key difference, prior works like Moulos (2019); Ariu et al. (2025) proceed via arguing that $\frac{\mathbb{E}[N_x(\tau_\alpha)]}{\mathbb{E}[\tau_\alpha]} = \pi_x + o(1)$ and can only obtain a lower bound asymptotically. Our approach allows us to characterize this $o(1)$ term exactly in terms of mixing properties of the problem instance.

# B. Proofs of the Results Appearing in Section 4

In this section, we first collect the necessary results, and subsequently apply them to complete the desired proof.

## B.1. Good Event Analysis

For $t \geq 1$, define the event

$$A_t := \left\{ t \inf_{P \in \mathcal{P}} \sum_{x \in [m]} \frac{N_x(t)}{t} \mathrm{D_{KL}}\left(\widehat{Q}_x(t), P_x\right) < \log(1/\alpha) + (m-1) \sum_{x \in [m]} \log\left(e\left(1 + \frac{N_x(t)}{m-1}\right)\right) \right\}. \tag{13}$$

Also, for any $\epsilon > 0$ and $t \geq 1$, define the *good set*

$$\mathcal{E}_\epsilon(t) := \left\{ \forall\, x \in [m], \ \left| \frac{N_x(t)}{t} - \pi_x \right| \leq \epsilon \right\} \cap \left\{ \|\widehat{Q}_t - Q\|_{1,\infty} \leq \epsilon \right\}. \tag{14}$$

Let $A_{t,\epsilon} := A_t \cap \mathcal{E}_\epsilon(t)$.

**Lemma B.1** (Good event analysis). *For any $\epsilon > 0$ and $t \geq 1$,*

$$A_{t,\epsilon} \subseteq \left\{ t \leq \frac{2C}{C(\epsilon)} \log\left(\frac{\log(1/\alpha) + D}{C(\epsilon)}\right) + \frac{\log(1/\alpha) + D}{C(\epsilon)} \right\},$$

*where:* $C := m(m-1)$, $D := 2m(m-1)$, *and* $C(\epsilon) := \inf_{Q':\|Q'-Q\|_{1,\infty}\leq\epsilon} \inf_{P \in \mathcal{P}} \sum_{x \in [m]} (\pi_x - \epsilon)\, \mathrm{D_{KL}}\left(Q'_x, P_x\right).$

*Proof.* Let

$$R := \log(1/\alpha) + (m-1) \sum_{x \in [m]} \log\left(e\left(1 + \frac{N_x(t)}{m-1}\right)\right), \quad L := t \inf_{P \in \mathcal{P}} \sum_{x \in [m]} \frac{N_x(t)}{t} \mathrm{D_{KL}}\left(\widehat{Q}_x(t), P_x\right).$$

Noting that $N_x(t) \leq t$ for all $x \in [m]$ and $t \geq 1$, we have

$$(m-1) \sum_{x \in [m]} \log \left( e \left( 1 + \frac{N_x(t)}{m-1} \right) \right) = m(m-1) + (m-1) \sum_{x \in [m]} \log \left( 1 + \frac{N_x(t)}{m-1} \right)$$

$$\leq m(m-1) + (m-1) \sum_{x \in [m]} \log \left( 1 + \frac{t}{m-1} \right)$$

$$\leq m(m-1) + (m-1) \sum_{x \in [m]} \log \left( t \left( 1 + \frac{1}{m-1} \right) \right)$$

$$\leq m(m-1) \log t + m(m-1) \left( 1 + \log \left( 1 + \frac{1}{m-1} \right) \right)$$

$$\leq C \log t + D,$$

where $C$ and $D$ are as defined in the statement of the lemma, and the last line follows by noting that $\log \left( 1 + \frac{1}{m-1} \right) \leq 1$ for all $m \geq 2$. We thus have

$$R \leq \log(1/\alpha) + C \log t + D.$$

On the good set $\mathcal{E}_\epsilon(t)$, noting that $N_x(t)/t \geq (\pi_x - \epsilon)$ for all $x \in [m]$, we have

$$L \geq t \inf_{P \in \mathcal{P}} \sum_{x \in [m]} (\pi_x - \epsilon) \, \mathrm{D_{KL}} \left( \widehat{Q}_x(t), P_x \right).$$

Let $C(\epsilon)$ be as defined in the statement of the lemma. Noting that $\|\widehat{Q}_t - Q\|_{1,\infty} \leq \epsilon$ on the set $\mathcal{E}_\epsilon(t)$, it follows that $tC(\epsilon) \leq L$. Thus, on $A_{t,\epsilon}$, we have

$$tC(\epsilon) \leq L < R \leq \log(1/\alpha) + C \log t + D.$$

Rearranging terms, we have

$$t \leq \frac{\log(1/\alpha) + D}{C(\epsilon)} + \frac{C \log t}{C(\epsilon)}$$

$$\leq \frac{2C}{C(\epsilon)} \log \left( \frac{\log(1/\alpha) + D}{C(\epsilon)} \right) + \frac{\log(1/\alpha) + D}{C(\epsilon)},$$

where the second line above follows from the technical Lemma F.1. This completes the proof. $\qquad \square$

**Lemma B.2** (Concentration of frequency vector to stationary distribution). *For an ergodic Markov chain with transition matrix $P$, associated unique stationary distribution $\pi$, and initial distribution $\mu$,*

$$\forall \, \epsilon > 0, \qquad \mathbb{P}_{P,\mu} \left[ \left| \frac{N_x(t)}{t} - \pi_x \right| > \epsilon \right] \leq \sqrt{2\Pi_\mu} \cdot \exp \left( - \frac{\epsilon^2 t^2 \gamma_{\mathrm{ps}}}{4(t + 1/\gamma_{\mathrm{ps}}) + 40\epsilon t} \right), \tag{15}$$

*where* $\Pi_\mu := \sum_{i \in [m]} \frac{\mu(i)^2}{\pi(i)} \leq \frac{1}{\pi^*}.$

*Proof of Lemma B.2.* From Proposition 3.15 of Paulin (2015), we have

$$\mathbb{P}_{P,\mu} \left[ \left| \frac{N_x(t)}{t} - \pi_x \right| > \epsilon \right] \leq \sqrt{\Pi_\mu} \left( \mathbb{P}_{P,\pi} \left[ \left| \frac{N_x(t)}{t} - \pi_x \right| > \epsilon \right] \right)^{1/2},$$

where $\mathbb{P}_{P,\pi}$ denotes the probability measure induced when the initial distribution is, in fact, the stationary distribution, and the state transitions are governed by the transition matrix $P$. We note that

$$\mathbb{P}_{P,\pi} \left[ \left| \frac{N_x(t)}{t} - \pi_x \right| > \epsilon \right] = \mathbb{P}_{P,\pi} \left[ |N_x(t) - t\pi_x| > t\epsilon \right].$$

Observe that:

1. $N_x(t) = \sum_{i=1}^{t} \mathbb{I}\{X_{i-1} = x\} = \sum_{i=0}^{t-1} \mathbb{I}\{X_i = x\}$.

2. For every $x \in [m]$,

$$\left| \mathbb{I}\{X = x\} - \mathbb{E}_\pi\left[ \mathbb{I}\{X = x\} \right] \right| \leq \max\{\pi_x, 1 - \pi_x\} \leq 1.$$

3. $\text{Var}_\pi(\mathbb{I}\{X = x\}) = \pi_x(1 - \pi_x) \leq 1/4$.

Applying Theorem 3.11 of Paulin (2015), we get

$$\mathbb{P}_{P,\pi}\left[ |N_x(t) - t\pi_x| > \epsilon t \right] \leq 2 \cdot \exp\left( -\frac{\epsilon^2 t^2 \gamma_{\text{ps}}}{2(t + 1/\gamma_{\text{ps}}) + 20\epsilon t} \right),$$

thereby establishing the desired result. □

## B.2. Concentration Analysis

**Lemma B.3** (Concentration Inequality for Empirical Transition Matrix (Wolfer & Kontorovich, 2019)). *For an ergodic Markov chain with finite state space $[m]$, unknown transition matrix $P$ and its associated unique stationary distribution $\pi$. For any initial distribution $\mu$, the empirical transition matrix $\widehat{P}(t)$, defined via $\widehat{P}_{x,y}(t) = \frac{N_{x,y}(t)}{N_x(t)}$ for all $x, y \in [m]$, satisfies*

$$\forall t \geq 1, \ \forall \epsilon > 0, \ \mathbb{P}_{P,\mu}\left[ \|P - \widehat{P}(t)\|_{1,\infty} > \epsilon \right] \leq 2m^2 \exp\left( -\frac{\epsilon^2 t \pi_*}{16m} \right) + m\sqrt{\Pi_\mu} \exp\left( -\frac{\gamma_{\text{ps}}(0.5t\pi_*)^2}{14t + 4/\gamma_{\text{ps}}} \right), \tag{16}$$

*where $\pi_* = \min_{x \in [m]} \pi_x$.*

*Proof of Lemma B.3.* From (Wolfer & Kontorovich, 2019, Theorem 1), for each $x \in [m]$ and $t \geq 1$, we have

$$\mathbb{P}_{P,\mu}\left[ \|\widehat{P}_{(x,\cdot)}(t) - P\|_1 > \epsilon, N_x(t) \in [0.5t\pi_x, 1.5t\pi_x] \right] \leq 2m \exp\left( -\frac{\epsilon^2 t \pi_*}{16m} \right). \tag{17}$$

Furthermore, from Lemma B.2 (which in turn follows from Paulin (2015) and is given as Lemma 6 in Wolfer & Kontorovich (2019)), for each $x \in [m]$, we have

$$\mathbb{P}_{P,\mu}\left[ N_x(t) \notin [0.5t\pi_x, 1.5t\pi_x] \right] \leq \sqrt{\Pi_\mu} \exp\left( -\frac{\gamma_{\text{ps}}(0.5t\pi_x)^2}{2(8(t + 1/\gamma_{\text{ps}})\pi_x(1 - \pi_x) + 10t\pi_x)} \right). \tag{18}$$

Let

$$E := \frac{\gamma_{\text{ps}}(0.5t\pi_x)^2}{2(8(t + 1/\gamma_{\text{ps}})\pi_x(1 - \pi_x) + 10t\pi_x)}.$$

To simplify $E$ further, we note that $\gamma_{\text{ps}}(0.5t\pi_x)^2 \geq \gamma_{\text{ps}}(0.5t\pi_*)^2$. The denominator term (say $D$) in the expression for $E$ may be upper bounded as

$$D = 2(8(t + 1/\gamma_{\text{ps}})\pi_x(1 - \pi_x) + 10t\pi_x) \leq 2(2(t + 1/\gamma_{\text{ps}}) + 10t) = 14t + 4/\gamma_{\text{ps}},$$

where the penultimate line follows by using the fact that $\pi_x \leq 1, \pi_x(1 - \pi_x) \leq 1/4$ for any $x \in [m]$. Therefore,

$$E \geq \frac{\gamma_{\text{ps}}(0.5t\pi_*)^2}{14t + 4/\gamma_{\text{ps}}},$$

from which it follows that

$$\mathbb{P}_{P,\mu}\left[ N_x(t) \notin [0.5t\pi_x, 1.5t\pi_x] \right] \leq \sqrt{\Pi_\mu} \exp\left( -\frac{\gamma_{\text{ps}}(0.5t\pi_*)^2}{14t + 4/\gamma_{\text{ps}}} \right). \tag{19}$$

An application of the union bound then yields

$$\mathbb{P}_{P,\mu}\left[ \|P - \widehat{P}(t)\|_{1,\infty} > \epsilon \right] \leq \sum_{x \in [m]} \mathbb{P}_{P,\mu}\left[ \|\widehat{P}_{(x,\cdot)}(t) - P\|_1 > \epsilon, N_x(t) \in [0.5t\pi_x, 1.5t\pi_x] \right]$$

$$+ \sum_{x \in [m]} \mathbb{P}_{P,\mu} \left[ N_x(t) \notin [0.5t\pi_x, 1.5t\pi_x] \right]$$

$$\leq 2m^2 \exp\left(-\frac{\epsilon^2 t\pi_*}{16m}\right) + m\sqrt{\Pi_\mu} \exp\left(-\frac{\gamma_{\mathrm{ps}}(0.5t\pi_*)^2}{14t + 4/\gamma_{\mathrm{ps}}}\right),$$

thereby establishing the desired result. $\qquad \square$

### B.3. Construction of Mixture Martingale

We use a mixture martingale construction from MDP literature (Jonsson et al., 2020; Al Marjani et al., 2021). We specialize Lemma 15 of Al Marjani et al. (2021) (in turn adapted from Proposition 1 of Jonsson et al. (2020)) to our setting. While exactly the same proof follows, we provide the details below for completeness. We collect some results to be used in the proof.

**Lemma B.4** (Lemma 3 in (Jonsson et al., 2020)). *For $q, p \in \Delta_m$ and $\lambda \in \mathbb{R}^{m-1} \times \{0\}$:*

$$\lambda^T q - \phi_q(\lambda) = D_{\mathrm{KL}}(q, p) - D_{\mathrm{KL}}\left(q, p^\lambda\right),$$

*where $p^\lambda = \nabla\phi_p(\lambda)$.*

**Lemma B.5** (Theorem 11.1.3 in (Cover & Thomas, 2012)). *Let $N \in \mathbb{N}, x \in \{0, 1, \ldots, N\}^k$ such that $\sum_{i \in [k]} x_i = N$ then:*

$$\binom{N}{x} := \frac{N!}{\prod_{i=1}^{k} x_i!} \leq \exp(NH(x/N)),$$

*where $H(x/N)$ is the entropy of the distribution over $k$ letters with corresponding probabilities $\{x_i/N\}_{i \in [k]}$.*

Now, we can prove the main result of this section. Note that for ease of reading, we prove Claim 1 and Claim 2 after the main proof.

**Lemma B.6.** *For $t \geq 1$ and $\alpha \in (0, 1)$, let $\beta(t, \alpha) := \log(1/\alpha) + (m-1) \sum_{x \in [m]} \log\left(e\left(1 + \frac{N_x(t)}{m-1}\right)\right)$. Then for all $\alpha \in (0, 1)$ and $P \in \mathcal{P}$,*

$$\mathbb{P}_P \left[ \exists t \geq 1 : \quad \sum_{x \in [m]} N_x(t) D(\widehat{P}(t)(x, \cdot), P(x, \cdot)) \geq \beta(t, \alpha) \right] \leq \alpha.$$

*Proof.* Let $|S| = m$. For the following definitions, $\lambda \in \mathbb{R}^{m-1} \times \{0\}, \lambda = (\lambda_1, \lambda_2, \ldots, \lambda_{m-1}, 0)$ and any distribution $p = (p_1, p_2, \ldots, p_m)$ over the $m$ states:

1. $\lambda^T p = \sum_{i=1}^{m-1} \lambda_i p_i$.

2. $\phi_p(\lambda) := \log(p_m + \sum_{i=1}^{m-1} p_i e^{\lambda_i})$.

**Constructing martingale for each state $x \in [m]$.**

Let $\widehat{p}_x(t)$ be the row corresponding to the state $x$ in the empirical transition matrix $\widehat{P}(t)$ and $\phi_x$ be the corresponding function for the true distribution for this row, i.e. $\phi_x(\lambda) = \phi_{P(x,\cdot)}(\lambda)$.

$$M_t^\lambda(x) = \exp(N_x(t)(\lambda^T \widehat{p}_x(t) - \phi_x(\lambda))). \tag{20}$$

*Claim 1.* $M_t^\lambda(x)$ is a martingale adapted to the sigma algebra generated sequence of states sampled from $P$ i.e. $\mathcal{F}_t = \sigma(X_0, X_1, \ldots, X_t)$.

Now, we define the mixture martingale given by the prior over $\lambda$ given by $\lambda_q = (\nabla\phi_x)^{-1}(q)$ where $q$ follows the Dirichlet distribution i.e. $q \sim Dir(1, 1, \ldots, 1)$.

$$M_t(x) := \int M_t^{\lambda_q}(x) \frac{\Gamma(m)}{\prod_{i=1}^m \Gamma(1)} \prod_{i=1}^m q_i dq$$

$$= \int \exp(N_x(t)(\lambda_q^T \widehat{p}_x(t) - \phi_x(\lambda_q))) \frac{\Gamma(m)}{\prod_{i=1}^m \Gamma(1)} \prod_{i=1}^m q_i dq$$

$$= \int \exp[N_x(t)(\mathrm{KL}(\widehat{p}_x(t), p_x) - \mathrm{KL}(\widehat{p}_x(t), q)](m-1)! \prod_{i=1}^m q_i dq \qquad \text{(Lemma B.4)}$$

$$= \exp[N_x(t)(\mathrm{KL}(\widehat{p}_x(t), p_x) + H(\widehat{p}_t(x)))](m-1)! \int \prod_{i=1}^m q_i^{1+N_x(t)\widehat{p}_{x,i}(t)} dq$$

$$(\because -\mathrm{KL}(\widehat{p}_x(t), q) = H(\widehat{p}_x(t)) + \sum \widehat{p}_x(t)(i) \log(q(i)))$$

$$= \exp[N_x(t)(\mathrm{KL}(\widehat{p}_x(t), p_x) + H(\widehat{p}_t(x)))] \frac{(m-1)! \prod_{i=1}^m \Gamma(1 + N_x(t)\widehat{p}_{x,i}(t))}{\Gamma(N_x(t) + m)}$$

$$= \exp[N_x(t)(\mathrm{KL}(\widehat{p}_x(t), p_x) + H(\widehat{p}_t(x)))] \frac{(m-1)! \prod_{i=1}^m (N_x(t)\widehat{p}_{x,i}(t))!}{(N_x(t) + m - 1)!}$$

$$= \exp[N_x(t)(\mathrm{KL}(\widehat{p}_x(t), p_x) + H(\widehat{p}_t(x)))] \frac{\prod_{i=1}^m (N_x(t)\widehat{p}_{x,i}(t))!}{N_x(t)!} \frac{(m-1)! N_x(t)!}{(N_x(t) + m - 1)!}$$

$$= \exp[N_x(t)(\mathrm{KL}(\widehat{p}_x(t), p_x) + H(\widehat{p}_t(x)))] \frac{1}{\binom{N_x(t)+m-1}{m-1}} \frac{1}{\binom{N_x(t)}{(N_x(t)\widehat{p}_{x,i}(t))_{i\in[k]}}},$$

where $\widehat{p}_{x,i}(t)$ is the $i^{\text{th}}$ component of the probability vector. Now, using Lemma B.5 to control the binomial coefficients:

$$M_t(x) \geq \exp[N_x(t)(\mathrm{KL}(\widehat{p}_x(t), p_x) + H(\widehat{p}_t(x)) - N_x(t)H(\widehat{p}_x(t))$$
$$- (N_x(t) + m - 1)H((m-1)/(N_x(t) + m - 1)]$$
$$= \exp[N_x(t)\mathrm{KL}(\widehat{p}_x(t), p_x) - (N_x(t) + m - 1)H((m-1)/(N_x(t) + m - 1)].$$

To upper bound the entropic term:

$$(N_x(t) + m - 1)H((m-1)/(N_x(t) + m - 1) = (m-1)\log\left(\frac{N_x(t) + m - 1}{m - 1}\right)$$
$$+ N_x(t)\log\left(\frac{N_x(t) + m - 1}{N_x(t)}\right)$$
$$= (m-1)\log\left(1 + \frac{N_x(t)}{m - 1}\right) + N_x(t)\log\left(1 + \frac{m - 1}{N_x(t)}\right)$$
$$\leq (m-1)\log\left(1 + \frac{N_x(t)}{m - 1}\right) + m - 1 \qquad (\because \log(1 + x) \leq x)$$
$$= (m-1)\log\left(e\left(1 + \frac{N_x(t)}{m - 1}\right)\right).$$

Thus, we have

$$M_t(x) \geq \exp\left[N_x(t)\mathrm{KL}(\widehat{p}_x(t), p_x) - (m-1)\log\left(e\left(1 + \frac{N_x(t)}{m - 1}\right)\right)\right].$$

**Product martingale over all states**

Finally, we take a product over all the states to get our final martingale.

$$M_t := \prod_{x\in[m]} M_t(x). \qquad (21)$$

Using the lower bound on each state martingale, we have

$$M_t \geq \exp\left[\sum_{x\in[m]} N_x(t)\mathrm{KL}(\widehat{p}_x(t), p_x) - (m-1)\sum_{x\in[m]}\log\left(e\left(1+\frac{N_x(t)}{m-1}\right)\right)\right]. \tag{22}$$

*Claim* 2. $M_t$ is a martingale adapted to the sigma algebra generated by the sequence of states sampled from the Markov Chain $P$.

Applying Ville's inequality:

$$\mathbb{P}\left[\exists t, M_t \geq 1/\alpha\right] \leq \alpha\mathbb{E}\left[M_0\right] = \alpha. \tag{23}$$

From (22) we have

$$\mathbb{P}\left[\exists t, \sum_{x\in[m]} N_x(t)\mathrm{KL}(\widehat{p}_x(t), p_x) \geq \log(1/\alpha) + (m-1)\sum_{x\in[m]}\log\left(e\left(1+\frac{N_x(t)}{m-1}\right)\right)\right]$$
$$\leq \mathbb{P}\left[\exists t, M_t \geq 1/\alpha\right]$$
$$\leq \alpha.$$

$\square$

*Proof of Claim 1.* Suppose $X_{t-1} \neq x$, then $N_x(t) = N_x(t-1)$ and $N_{x,y}(t) = N_{x,y}(t-1)\forall y \in [m]$, therefore $\widehat{p}_x(t) = \widehat{p}_x(t-1)$. Thus, since none of the data dependent parameters change: $M_t^\lambda(x) = M_{t-1}^\lambda(x)$.

If $X_{t-1} = x$, then:

$$\mathbb{E}\left[\lambda^T\widehat{p}_x(t)|X_0,\ldots,X_{t-1} - x\right] = \mathbb{E}\left[\lambda^T\widehat{p}_x(t)|X_0,\ldots,X_{t-1} = x\right]$$
$$= \mathbb{E}_{X_t\sim P(x,\cdot)}\left[\lambda^T\left(\frac{N_x(t-1)\widehat{p}_x(t-1) + \bar{X}_t}{N_x(t-1)+1}\right)\right]$$
$$= \frac{1}{N_x(t-1)+1}\mathbb{E}_{X_t\sim P(x,\cdot)}\left[\lambda^T\left(N_x(t-1)\widehat{p}_x(t-1) + \bar{X}_t\right)\right],$$

where $\bar{X}_t \in \{0,1\}^m$ is the one-hot encoding of the sample $X_t$ i.e. $\bar{X}_t(i) = \mathbb{I}\{X_t = i\}\ \forall i \in [m]$. Now, we have:

$$\mathbb{E}\left[M_t^\lambda(x)|X_0, X_1,\ldots,X_{t-1} = x\right] = \mathbb{E}\left[\exp(N_x(t)(\lambda^T\widehat{p}_x(t) - \phi_x(\lambda))|X_0, X_1,\ldots,X_{t-1} = x\right]$$
$$= \mathbb{E}_{X_t\sim P(x,\cdot)}\left[\exp\left((N_x(t-1)+1)(\lambda^T\left(\frac{N_x(t-1)\widehat{p}_x(t-1) + \bar{X}_t}{N_x(t-1)+1}\right) - \phi_x(\lambda))\right)|\ldots\right]$$
$$= \mathbb{E}_{X_t\sim P(x,\cdot)}\left[\exp\left((\lambda^T\left(N_x(t-1)\widehat{p}_x(t-1) + \bar{X}_t\right) - (N_x(t-1)+1)\phi_x(\lambda))\right)|\ldots\right]$$
$$= M_{t-1}^\lambda(x)\mathbb{E}_{X_t\sim P(x,\cdot)}\left[\exp(\lambda^T\bar{X}_t - \phi_x(\lambda))|\ldots,\right]$$
$$= M_{t-1}^\lambda(x).$$

$\square$

*Proof of Claim 2.* Fix some state $x \in [m]$. Then observe:

$$\mathbb{E}\left[M_t|X_0, X_1,..X_{t-1} = x\right] = \mathbb{E}\left[M_t(x)\prod_{y\neq x}M_t(y)|X_0, X_1,..X_{t-1} = x\right]$$
$$= \mathbb{E}\left[M_t(x)\prod_{y\neq x}M_{t-1}(y)|X_0, X_1,..X_{t-1} = x\right]$$

$$= \prod_{y \neq x} M_{t-1}(y) \mathbb{E}\left[M_t(x)|X_0, X_1, ..X_{t-1} = x\right]$$

$$= M_{t-1},$$

where the second to last equality follows from observing the conditional independence of $M_t(x)$ and $\{M_{t-1}(y) : y \neq x\}$. Since this holds for each state in $S$, we have the martingale property by applying the tower rule. □

## B.4. Continuity Properties

**Lemma B.7.** *Let* $C(\epsilon) := \inf_{Q':\|Q'-Q\|_{1,\infty} \leq \epsilon} \inf_{P \in \mathcal{P}} \sum_{x \in [m]} (\pi_x - \epsilon) D_{\mathrm{KL}}\left(Q'_x, P_x\right)$, *then:*

$$\liminf_{\epsilon \downarrow 0} C(\epsilon) \geq D_{\mathcal{M}}^{\inf}(Q, \mathcal{P}).$$

*Proof.* To show the result, it suffices to show the lower semi-continuity of $C$, as done in Lemma B.19, since that implies $\liminf_{\epsilon \to 0} C(\epsilon) \geq C(0) = D_{\mathcal{M}}^{\inf}(Q, \mathcal{P})$. □

To show Lemma B.19, we require some tools from optimization theory. For convenience, we collect some basic definitions and results before proceeding to the formal arguments.

### B.4.1. PRELIMINARIES

We recall some basic tools from Chapter 6 of (Berge, 1997).

**Definition B.8** (Correspondence)**.** Let $X, Y$ be topological spaces, a set valued function $f : X \to 2^Y$ is said to be a correspondence, i.e. for each $x \in X, f(x) \subseteq Y$.

**Definition B.9** (Lower semi-continuity of a correspondence)**.** Let $X, Y$ be topological spaces. A correspondence $f : X \to 2^Y$ is said to be lower semi-continuous at $x_0 \in X$, if for each open set $G$ intersecting $f(x_0)$, there exists a neighbourhood $U(x_0)$ of $x_0$ such that: $x \in U(x_0) \implies f(x) \bigcap G \neq \varnothing$.

**Definition B.10** (Upper semi-continuity of a correspondence)**.** Let $X, Y$ be topological spaces. A correspondence $f : X \to 2^Y$ is said to be upper semi-continuous at $x_0 \in X$, if for each open set $G$ containing $f(x_0)$, there exists a neighbourhood $U(x_0)$ of $x_0$ such that: $x \in U(x_0) \implies f(x) \subseteq G$.

Observe that if correspondence has co-domain is over the reals and is singleton valued, i.e. a real-valued function, these definitions coincide with the more familiar definitions.

**Definition B.11** (Upper semi-continuity of real valued functions)**.** Let $X$ be a topological space, a function $f : X \to \mathbb{R} \cup \{-\infty, \infty\}$ is said to be upper semi-continuous at $x_0 \in X$ if for every $y$ such that $f(x_0) < y$, there exists a neighbourhood $U$ of $x_0$ such that $f(x) < y \; \forall x \in U$.

**Definition B.12** (Lower semi-continuity of real valued functions)**.** Let $X$ be a topological space, a function $f : X \to \mathbb{R} \cup \{-\infty, \infty\}$ is said to be lower semi-continuous at $x_0 \in X$ if for every $y$ such that $f(x_0) > y$, there exists a neighbourhood $U$ of $x_0$ such that $f(x) > y \; \forall x \in U$.

*Remark* B.13. A correspondence is called lower (resp. upper) semi-continuous if it is lower (resp. upper) semi-continuous at each point in its domain.

**Theorem B.14** (Maximum Theorem, Berge (1997, pg. 116))**.** *If* $\phi : X \times Y \to \mathbb{R}$ *is an upper semi-continuous function and* $\Gamma : X \to 2^Y$ *is an upper semi-continuous, compact-valued correspondence such that, for each* $x, \Gamma(x) \neq \varnothing$*, then* $M : X \to \mathbb{R}$ *defined as:*

$$M(x) := \max\{\phi(x, y) : y \in \Gamma(x)\},$$

*is upper semi-continuous.*

We require the minimizing version of the theorem, which we state here for completeness.

**Corollary B.15** (Minimum Theorem)**.** *If* $\phi : X \times Y \to \mathbb{R}$ *is a lower semi-continuous function and* $\Gamma : X \to 2^Y$ *is an upper semi-continuous, compact-valued correspondence such that, for each* $x, \Gamma(x) \neq \varnothing$*, then* $M : X \to \mathbb{R}$ *defined as:*

$$M(x) := \min\{\phi(x, y) : y \in \Gamma(x)\},$$

*is lower semi-continuous.*

*Proof.* Observe that if $\phi$ is lower semi-continuous, then $-\phi$ is upper semi-continuous. Applying Maximum theorem, we have that $M'(x) = \max\{-\phi(x,y) : y \in \Gamma(x)\} = -\min \phi(x,y) : y \in \Gamma(x)$ is upper semi-continuous, thus, $M(x) = \min\{\phi(x,y) : y \in \Gamma(x)\}$ is lower semi-continuous. $\square$

### B.4.2. RESULTS

Let $\mathcal{M}$ be the set of $m \times m$ stochastic matrices. We show the following lemmas before showing the requisite lower semi-continuity of $C(\epsilon)$.

**Lemma B.16** (Compactness of $\mathcal{M}$). *Let $\mathcal{M} := \{A \in \mathbb{R}^{m \times m} : A$ is stochastic i.e. $\sum_{j \in [m]} A(i,j) = 1 \; \forall i \in [m], A(i,j) \geq 0 \; \forall i,j\}$. $\mathcal{M} \subset \mathbb{R}^{n \times n}$ is compact under the standard topology.*

*Proof.* We will show that $\mathcal{M}$ is closed and bounded in the topology generated by $\|\cdot\|_{1,\infty}$. Observe that for each $P \in \mathcal{M}, \|P\|_{1,\infty} = 1$, thus the set is bounded. Consider $\{P_n\} \to P^*$. Fix any row $i$ and observe:

$$\sum_{j \in [m]} P^*(i,j) = \sum_{j \in [m]} \lim_{n \to \infty} P_n(i,j) = \lim_{n \to \infty} \sum_{j \in [m]} P_n(i,j) = 1.$$

Further, observe that $P^n(i,j) \geq 0$ for each entry thus, $P^*(i,j) \geq 0$. Thus, since this set is a closed and bounded subset of a finite dimensional normed space, i.e. $\mathbb{R}^{m \times m}$, we have compactness by Kreyszig (1978, Theorem 2.5-3). $\square$

**Lemma B.17** (Upper semi-continuity of correspondence). *The correspondence $\Gamma(\epsilon) := \{Q' \in \mathcal{M} : \|Q' - Q\|_{1,\infty} \leq \epsilon\}$ is upper semi-continuous and compact valued.*

*Proof.* Consider $\Gamma : \mathbb{R}_{++} \to 2^{\mathcal{M}}$. Let $\epsilon_0 > 0$ then, $\Gamma(\epsilon_0) = \{Q' \in \mathcal{M} : \sup_{i \in [m]} \|Q'(i,\cdot) - Q(i,\cdot)\|_1 \leq \epsilon_0\}$. Compactness of $\Gamma(\epsilon_0)$ follows by observing it is a closed and bounded subset of a finite dimensional normed vector space (Kreyszig, 1978, Theorem 2.5-3).

To show upper semi-continuity of $\Gamma$, we demonstrate upper semi-continuity at any $\epsilon_0 \in \mathbb{R}_{++}$. Let $G$ be an arbitrary open set in $\mathcal{M}$ such that $\Gamma(\epsilon_0) \subseteq G$. We must find a neighborhood $(\epsilon_0 - \delta, \epsilon_0 + \delta)$ such that for all $\epsilon$ in this neighborhood, $\Gamma(\epsilon) \subseteq G$.

We proceed by contradiction. Suppose no such $\delta$ exists. Then, for every $n \in \mathbb{N}$, we can find an $\epsilon_n$ such that $|\epsilon_n - \epsilon_0| < 1/n$ and yet $\Gamma(\epsilon_n) \not\subseteq G$. This implies that for each $n$, there exists a matrix $Q'_n \in \Gamma(\epsilon_n)$ such that $Q'_n \notin G$. Consider the sequence $(Q'_n)_{n \in \mathbb{N}}$.

By definition of $\Gamma(\epsilon_n)$, we have $\|Q'_n - Q\|_{1,\infty} \leq \epsilon_n$. Since the space of stochastic matrices $\mathcal{M}$ is compact (Lemma B.16), the sequence $(Q'_n)$ must have a convergent subsequence $(Q'_{n_k})$ that converges to some limit point $Q^* \in \mathcal{M}$.

Taking the limit of the inequality along this subsequence:

$$\|Q^* - Q\|_{1,\infty} \leq \lim_{k \to \infty} \|Q'_{n_k} - Q\|_{1,\infty} \leq \lim_{k \to \infty} \epsilon_{n_k} = \epsilon_0,$$

where we apply the triangle inequality to $\|Q^* - Q'_{n_k} + Q'_{n_k} - Q\|_{1,\infty}$ and then take the limit as $k \to \infty$. Therefore, $Q^* \in \Gamma(\epsilon_0)$. Since $\Gamma(\epsilon_0) \subseteq G$, it must be that $Q^* \in G$. However, $G$ is an open set. Since the subsequence converges to $Q^*$ (which is inside $G$), the tail of the subsequence must eventually enter $G$. This contradicts the assumption that $Q'_n \notin G$ for all $n$. Thus, our assumption was false. A $\delta$ must exist, proving that $\Gamma$ is upper semi-continuous.

$\square$

**Lemma B.18** (Joint lower semi-continuity). *Let $\phi : \mathbb{R}_{++} \times \mathcal{M} \times \mathcal{P} \to \mathbb{R} \bigcup \{+\infty\}$ be given as*

$$\phi(\epsilon, Q', P) := \sum_{x \in [m]} (\pi_x - \epsilon) D_{\mathrm{KL}} \left(Q'_x, P_x\right).$$

*This function is jointly lower semi-continuous in its arguments for $\epsilon < \pi_*$.*

*Proof.* Observe that convergence in the $\ell_{1,\infty}$ norm implies convergence in total variation for each row probability vector. Further, recall that convergence in total variation implies weak convergence. Thus, $Q_n \overset{\ell_{1,\infty}}{\to} Q \implies Q_n(x, \cdot) \overset{w}{\to} Q(x, \cdot) \, \forall x \in [m]$.

First, it is well known that KL divergence $(p, q) \mapsto \mathrm{KL}(p, q)$ is lower-semi-continuous in the topology of weak convergence i.e. for $p_n \overset{w}{\to} p, q_n \overset{w}{\to} q$, we have $\liminf_{n \to \infty} \mathrm{KL}(p_n, q_n) \geq \mathrm{KL}(p, q)$ (see Theorem 4.9 in (Polyanskiy & Wu, 2025), originally shown by (Posner, 1975)). Since we're in finite dimensions, lower semi-continuity in the topology of weak convergence is equivalent to lower semi-continuity in the topology induced by the norm $\ell_{1,\infty}$ (Kreyszig, 1978, Theorem 4.8-4, part c).

Second, consider the weight map $w_x(\epsilon) := \pi_x - \epsilon$ is continuous and positive for $\epsilon \leq \pi^* := \min_{x \in [m]} \pi_x$ (recall that $\pi^*$ is positive by Perron Frobenius theorem). Since the product of a positive continuous function and a non-negative lower semi-continuous function is lower semi-continuous, each term $(\pi_x - \epsilon)\mathrm{D}_{\mathrm{KL}}((, Q)'_x \| P_x)$ is jointly lower semi-continuous.

Finally, since the sum of lower semi-continuous functions is lower semi-continuous, we have the result.

$\square$

**Lemma B.19.** $C(\epsilon) := \inf_{Q': \|Q' - Q\|_{1,\infty} \leq \epsilon} \inf_{P \in \mathcal{P}} \sum_{x \in [m]} (\pi_x - \epsilon)\mathrm{D}_{\mathrm{KL}}(Q'_x, P_x)$ *is lower semi-continuous.*

*Proof.* We proceed by applying the Minimum theorem twice.

First, consider the lower semi-continuous map $\phi(\epsilon, Q', P) := \sum_{x \in [m]} (\pi_x - \epsilon)\mathrm{D}_{\mathrm{KL}}(Q'_x, P_x)$ with the constant correspondence given by $\Gamma_1(P) \mapsto \mathcal{P} \, \forall P \in \mathcal{P}$. Clearly, this is compact valued (since $\mathcal{P}$ is compact) and continuous. Therefore, from Corollary B.15, we have that $\phi'(\epsilon, Q') := \min_{P \in \mathcal{P}} \phi(\epsilon, Q', P)$ is lower semi-continuous.

Secondly, consider this lower semi-continuous map and the compact valued, upper semi-continuous correspondence $\epsilon \mapsto \{Q' \in \mathcal{M} : \|Q' - Q\|_{1,\infty} \leq \epsilon\}$ (see Lemma B.17). Applying Corollary B.15, we have that $C(\epsilon)$ is lower semi-continuous.

$\square$

### B.5. Proof of Optimality

*Proof of Theorem 4.1.*
To show $\alpha$-correctness, we use the mixture martingale introduced in Section B.3.

$$\mathbb{P}_P[\tau_\alpha < \infty] = \mathbb{P}_P\left[\exists t, \inf_{P' \in \mathcal{P}} \sum_{x \in [m]} N_x(t)D(\widehat{P}(t)(x, \cdot), P'(x, \cdot)) \geq \beta(t, \alpha)\right]$$

$$\leq \mathbb{P}_P\left[\exists t, \sum_{x \in [m]} N_x(t)D(\widehat{P}(t)(x, \cdot), P(x, \cdot)) \geq \beta(t, \alpha)\right]$$

$$\leq \alpha. \qquad \text{(using Lemma B.6)}$$

From Lemma B.3 and Lemma B.2, we have that for any $t \geq 1$:

$$\mathbb{P}\left[\mathcal{E}_\epsilon^{\complement}(t)\right] \leq \sqrt{2\Pi_\mu} \cdot \exp\left(-\frac{\epsilon^2 t^2 \gamma_{\mathrm{ps}}}{4(t + 1/\gamma_{\mathrm{ps}}) + 40\epsilon t}\right) + 2m^2 \exp\left(-\frac{\epsilon^2 t \pi_*}{16m}\right)$$
$$+ m\sqrt{\Pi_\mu} \exp\left(-\frac{\gamma_{\mathrm{ps}}(0.5t\pi_*)^2}{14t + 4/\gamma_{\mathrm{ps}}}\right), \tag{24}$$

which is summable across time, i.e. $\sum_{t=1}^{\infty} \mathbb{P}\left[\mathcal{E}_\epsilon^{\complement}(t)\right] < \infty$.

From Lemma B.3, we have that $\|\widehat{Q}_t - Q\|_{1,\infty} \to 0$.

Pick $\epsilon > 0, \epsilon < \pi^*$, where $\pi^* := \min_{j \in [m]} \pi_j$. Define $A_t$ and $\mathcal{E}_\epsilon$ as in the previous sections.

$$\mathbb{E}_Q\left[\tau_\alpha\right] = \sum_{t=1}^{\infty} \mathbb{P}\left[\tau_\alpha \geq t\right]$$

$$\leq \sum_{t=1}^{\infty} \mathbb{P}\left[L_t < \beta(t, \alpha)\right]$$

$$= \sum_{t=1}^{\infty} \mathbb{P}\left[A_t\right]$$

$$= \sum_{t=1}^{\infty} \mathbb{P}\left[A_t \cap \mathcal{E}_\epsilon(t)\right] + \sum_{t=1}^{\infty} \mathbb{P}\left[\mathcal{E}_\epsilon^{\complement}(t)\right]$$

$$\leq \sum_{t=1}^{\infty} \mathbb{P}\left[t \leq \frac{2C}{C(\epsilon)}\log\left(\frac{\log(1/\alpha) + D}{C(\epsilon)}\right) + \frac{\log(1/\alpha) + D}{C(\epsilon)}\right] + \sum_{t=1}^{\infty} \mathbb{P}\left[\mathcal{E}_\epsilon^{\complement}\right]$$

$$\leq \frac{2C}{C(\epsilon)}\log\left(\frac{\log(1/\alpha) + D}{C(\epsilon)}\right) + \frac{\log(1/\alpha) + D}{C(\epsilon)} + \sum_{t=1}^{\infty} \mathbb{P}\left[\mathcal{E}_\epsilon^{\complement}(t)\right].$$

Dividing by $\log(1/\alpha)$ and taking the limit as $\alpha \to 0$, we have

$$\lim_{\alpha \to 0} \frac{\mathbb{E}_Q\left[\tau_\alpha\right]}{\log(1/\alpha)} = \frac{1}{C(\epsilon)}.$$

Letting $\epsilon \to 0$ and using the lower semi-continuity of $C(\epsilon)$, shown in Lemma B.7, we have the result.

$\square$

## C. Extension to Two-Sided Sequential Testing

In this section, we provide a proof of Theorem 4.4, stated in Section 4.2.

*Proof of Theorem 4.4.*
**Lower Bound:** We remark that the lower bound follows very similarly to the one-sided case, with the only change coming in the event chosen for the data processing inequality. Suppose the alternate hypothesis is true and the data is generated by some $Q \in \mathcal{Q}$ and we fix some $P \in \mathcal{P}$ such that $Q \ll P$. Following the proof of Theorem 3.3, we have

$$\mathbb{E}_Q\left[\tau_{\alpha,\beta}\right] D_{\mathcal{M}}(Q, P) + C_\tau^Q \geq d(Q(\mathcal{E}), P(\mathcal{E})) \,\forall\, \mathcal{E} \in \mathcal{F}_\tau, \tag{25}$$

where $C_\tau^Q = \mathbb{E}_Q[\omega(X_0) - \omega(X_{\tau_{\alpha,\beta}})]$.

Choosing $\mathcal{E} = \{D(\tau_{\alpha,\beta}) = 0\}$. Then, by the $(\alpha, \beta)$ correctness, we must have: $P(\mathcal{E}) \geq 1 - \alpha, Q(\mathcal{E}) \leq \beta$ (observe that this implies $P(\mathcal{E}) \geq 0.5 \geq Q(\mathcal{E})$). From the monotonicity properties of the Bernoulli KL divergence, we have

$$d(Q(\mathcal{E}), P(\mathcal{E})) \geq d(Q(\mathcal{E}), 1 - \alpha) \geq d(\beta, 1 - \alpha).$$

Combining the equations we have the result. Following the proof of Theorem 3.3, we bound the excess Poisson terms using Proposition 3.1 to allow us to safely maximize over $P$.

A similar proof follows for the case where the null is true.

**Construction of two-sided test:** The basic idea for the construction is to run two one-sided tests in parallel: one testing $\mathcal{P}$ vs $\mathcal{Q}$ and the other testing $\mathcal{Q}$ vs $\mathcal{P}$. We can choose the parameters such that the required error levels are met.

Let $\tau_\alpha^{\mathcal{P}}$ be an $\alpha$-correct, power-one test for testing $\mathcal{P}$ against $\mathcal{Q}$. By definition, we have

$$\mathbb{P}_P\left[\tau_\alpha^{\mathcal{P}} < \infty\right] \leq \alpha, \mathbb{P}_Q\left[\tau_\alpha^{\mathcal{P}} < \infty\right] = 1, \quad \forall\, P \in \mathcal{P}, Q \in \mathcal{Q}. \tag{26}$$

Now, let $\tau_\beta^{\mathcal{Q}}$ be a level $\beta$, power-one test for testing $\mathcal{Q}$ against $\mathcal{P}$. By definition, we have

$$\mathbb{P}_Q\left[\tau_\beta^{\mathcal{Q}} < \infty\right] \leq \beta, \mathbb{P}_P\left[\tau_\beta^{\mathcal{Q}} < \infty\right] = 1 \quad \forall\, P \in \mathcal{P}, Q \in \mathcal{Q}. \tag{27}$$

Define the parallel test: $\tau_{\alpha,\beta} = \min\{\tau_\alpha^{\mathcal{P}}, \tau_\beta^{\mathcal{Q}}\}$ with the decision rule, $D(\tau_{\alpha,\beta}) = 1$ if $\tau_\alpha^{\mathcal{P}} \leq \tau_\beta^{\mathcal{Q}}$, and $D(\tau_{\alpha,\beta}) = 0$ otherwise (since $\tau_\alpha^{\mathcal{P}} < \infty$ corresponds to choosing $H_1$ and vice-versa). We show that this test satisfies the correctness properties. Let $P \in \mathcal{P}$, then:

$$\mathbb{P}_P\left[D(\tau_{\alpha,\beta}) = 1\right] = \mathbb{P}_P\left[\tau_\alpha^{\mathcal{P}} \leq \tau_\beta^{\mathcal{Q}}\right] \leq \mathbb{P}_P\left[\tau_\alpha^{\mathcal{P}} < \infty\right] \leq \alpha.$$

Similarly, for any $Q \in \mathcal{Q}$, $\mathbb{P}_Q\left[D(\tau_{\alpha,\beta}) = 0\right] \leq \beta$.

We remark that this construction yields asymptotically optimal tests in the i.i.d. setting as well, as shown by Lorden (1976, Theorem 1). We show that the same holds for Markovian data.

**Asymptotic optimality:** Now, we prove the asymptotic optimality of $\tau_{\alpha,\beta}$ described above as $\alpha, \beta \to 0$.

Suppose the data is generated by some $Q \in \mathcal{P}$, the lower bound states that:

$$\mathbb{E}_Q\left[\tau_{\beta,\alpha}\right] \geq \left(\frac{d(\beta, 1-\alpha)}{D_{\mathcal{M}}^{\inf}(Q, \mathcal{P})} - \frac{2C_Q}{\pi_Q^*}\right)^+.$$

Dividing by $\log(1/\alpha)$ on either side and taking the limit $\alpha, \beta \to 0$, we obtain:

$$\liminf_{\alpha,\beta \to 0} \frac{\mathbb{E}_Q\left[\tau_{\alpha,\beta}\right]}{\log(1/\alpha)} \geq \lim_{\alpha,\beta \to 0} \frac{d(\beta, 1-\alpha)}{\log(1/\alpha) D_{\mathcal{M}}^{\inf}(Q, \mathcal{P})} = \frac{1}{D_{\mathcal{M}}^{\inf}(Q, \mathcal{P})}, \tag{28}$$

where the equality follows from Lemma F.2.

Now, we consider the test defined above, i.e. $\tau_{\alpha,\beta} = \min\{\tau_\alpha^{\mathcal{P}}, \tau_\beta^{\mathcal{Q}}\}$, where $\tau_\alpha^{\mathcal{P}}$ is the $\alpha-$correct, power-one test for testing compact $\mathcal{P}$ against $\mathcal{Q}$ given by Algorithm 1. Similarly, $\tau_\beta^{\mathcal{Q}}$ is the $\alpha-$correct, power-one test for testing compact $\mathcal{Q}$ against $\mathcal{P}$ given by Algorithm 1.

$$\mathbb{E}_Q\left[\tau_{\alpha,\beta}\right] = \mathbb{E}_Q\left[\min\{\tau_\alpha^{\mathcal{P}}, \tau_\beta^{\mathcal{Q}}\}\right] \leq \mathbb{E}_Q\left[\tau_\alpha^{\mathcal{P}}\right].$$

We divide by $\log(1/\alpha)$ take the limit as $\alpha, \beta \to 0$. Optimality follows by comparing this with (28). A similar proof holds when the data is generated by some ergodic $P \in \mathcal{P}$. $\qquad\square$

# D. Computationally Tractable Lower Bound

*Proof of Proposition 4.3.* Fix some $g : [m] \to \mathbb{R}$ such that it is not constant across states. Let ergodic $P, Q \in \mathcal{M}$ such that they have the stationary distributions $\pi_P, \pi_Q$. Further, suppose $\omega_{P,g}$ is a solution to the following PE:

$$(I - P)\omega_P = g - \mathbb{E}_{\pi_P}[g]. \tag{29}$$

Recall that such a solution always exists for finite state, ergodic chains and since $g$ is not constant, $\omega \neq \mathbf{0}$.

Now, observe:

$$\begin{aligned}
\mathbb{E}_{\pi_Q}[g] - \mathbb{E}_{\pi_P}[g] &= \mathbb{E}_{\pi_Q}\left[g - \mathbb{E}_{\pi_P}[g]\right] \\
&= \mathbb{E}_{\pi_Q}\left[(I - P)\omega_{P,g}\right] & \text{(from (29))} \\
&= \mathbb{E}_{\pi_Q}[\omega_{P,g}] - \mathbb{E}_{\pi_Q}[P\omega_{P,g}] \\
&= \mathbb{E}_{\pi_Q}[Q\omega_{P,g}] - \mathbb{E}_{\pi_Q}[P\omega_{P,g}] & (\because \pi_Q \omega_{P,g} = \pi_Q Q \omega_{P,g}) \\
&= \mathbb{E}_{\pi_Q}\left[(Q - P)\omega_{P,g}\right]
\end{aligned}$$

$$= \sum_{i \in [m]} \pi_Q(i) \left( \mathbb{E}_{Q(x,\cdot)} [\omega_{P,g}] - \mathbb{E}_{P(x,\cdot)} [\omega_{P,g}] \right).$$

From the variational form of Total Variation distance (Polyanskiy & Wu, 2025, Theorem 7.7, (a)), for any $Q_x, P_x \in \Delta_m$:

$$TV(Q_x, P_x) = \frac{1}{2} \sup_{f:\|f\|_\infty \leq 1} |\mathbb{E}_{Q_x} [f] - \mathbb{E}_{P_x} [f]|.$$

As a consequence, for any $f : [m] \to \mathbb{R}$, we have:

$$TV(Q_x, P_x) \geq \frac{1}{2\|f\|_\infty} |\mathbb{E}_{Q_x} [f] - \mathbb{E}_{P_x} [f]|.$$

Combining this with Pinkser's inequality, we have

$$\mathrm{D}_{\mathrm{KL}} (Q_x, P_x) \geq \frac{1}{2\|f\|_\infty^2} (\mathbb{E}_{Q_x} [f] - \mathbb{E}_{P_x} [f])^2.$$

From our previous analysis, we have

$$
\begin{aligned}
|\mathbb{E}_{\pi_Q} [g] - \mathbb{E}_{\pi_P} [g]| &\leq \sum_{i \in [m]} \pi_Q(i) \left| \mathbb{E}_{Q(x,\cdot)} [\omega_{P,g}] - \mathbb{E}_{P(x,\cdot)} [\omega_{P,g}] \right| \\
&\leq \sum_{i \in [m]} \pi_Q(i) \sqrt{2\|\omega_{P,g}\|_\infty^2 \mathrm{D}_{\mathrm{KL}} (Q(i,\cdot), P(i,\cdot))} \\
&= \sqrt{2}\|\omega_{P,g}\|_\infty \sum_{i \in [m]} \pi_Q(i) \sqrt{\mathrm{D}_{\mathrm{KL}} (Q(i,\cdot), P(i,\cdot))} \\
&\leq \sqrt{2}\|\omega_{P,g}\|_\infty \sqrt{\sum_{i \in [m]} \pi_Q(i) \mathrm{D}_{\mathrm{KL}} (Q(i,\cdot), P(i,\cdot))}. \qquad \text{(Jensen's inequality)}
\end{aligned}
$$

Thus, we have

$$D_{\mathcal{M}}(Q, P) \geq \frac{1}{2\|\omega_{P,g}\|_\infty^2} (\mathbb{E}_{\pi_Q} [g] - \mathbb{E}_{\pi_P} [g])^2. \tag{30}$$

$\square$

**Proposition D.1.** *For ergodic transition matrices $Q, P \in \mathcal{M}$ with stationary distributions $\pi_Q, \pi_P$ respectively, the set of maximizers of the optimization problem:*

$$\sup_{g:[m]\to\mathbb{R}} \frac{\left( \mathbb{E}_{\pi_Q} [g] - \mathbb{E}_{\pi_P} [g] \right)^2}{2\|\omega_{P,g}\|_\infty^2},$$

*is the set $\{c\, g^\star + b\mathbf{1} : c \neq 0, b \in \mathbb{R}\}$, where $g^* = (I - P)\omega^*$. The vector $\omega^* \in [-1,1]^m$ is defined coordinate-wise for a threshold $\mu^*$ as:*

$$\omega_i^* = \begin{cases} +1, & a_i > \mu^* \pi_P(i) \\ -1, & a_i < \mu^* \pi_P(i) \\ t_i, & a_i = \mu^* \pi_P(i) \end{cases}$$

*where $\mu^* \in \mathbb{R}$ is chosen such that the probability masses strictly above and strictly below the threshold are bounded by $1/2$:*

$$\sum_{i:a_i>\mu^*\pi_P(i)} \pi_P(i) \leq \frac{1}{2} \quad \text{and} \quad \sum_{i:a_i<\mu^*\pi_P(i)} \pi_P(i) \leq \frac{1}{2},$$

*and $t_i \in [-1, 1]$ are chosen to satisfy the constraint $\langle \pi_P, \omega^* \rangle = 0$.*

*Proof.* Fix an ergodic transition matrix $P$ on $[m]$ with stationary distribution $\pi_P$. Recall the Poisson solution $\omega_{P,g} \in \mathbb{R}^m$ as in (2) for $(P, g)$. Given ergodic $Q \in \mathcal{M}$ with stationary distribution $\pi_Q$, consider the functional

$$\psi(g) := \frac{1}{2\|\omega_{P,g}\|_\infty^2} \left( \langle \pi_Q, g \rangle - \langle \pi_P, g \rangle \right)^2 = \frac{1}{2\|\omega_{P,g}\|_\infty^2} \langle \pi_Q - \pi_P, g \rangle^2. \tag{31}$$

The map $g \mapsto \psi(g)$ is invariant to adding constants $(g \mapsto g + c\mathbf{1})$ and to scaling $(g \mapsto cg)$. Hence, without loss of generality, restrict to $\langle \pi_P, g \rangle = 0$ and optimize over directions.

On the subspace $\{g : \langle \pi_P, g \rangle = 0\}$, (1) reduces to $(I - P)\omega = g$ with $\langle \pi_P, \omega \rangle = 0$, so we can reparameterize $g = (I - P)\omega$. For a function of this form, we observe that the solution as in (2) is exactly equal to $\omega$, i.e. $\omega_{P,g} = \omega$. Therefore, we use the two interchangeably throughout this section.

Using $\langle \pi_P, (I - P)\omega \rangle = 0$, we obtain

$$\langle \pi_Q - \pi_P, g \rangle = \langle \pi_Q, g \rangle = \langle \pi_Q, (I - P)\omega \rangle = \omega^\top (I - P^\top)\pi_Q = \langle a, \omega \rangle, \tag{32}$$

where $a := (I - P^\top)\pi_Q$. Following this, we can rewrite the inner supremum in (6) as:

$$\sup_{\substack{g \in \mathbb{R}^m: \\ \langle \pi_P, g \rangle = 0}} \psi(g) = \sup_{\substack{g \in \mathbb{R}^m: \\ \langle \pi_P, g \rangle = 0}} \frac{1}{2\|\omega_{P,g}\|_\infty^2} \langle \pi_Q, g \rangle^2 = \sup_{\substack{g \in \mathbb{R}^m: \\ \langle \pi_P, g \rangle = 0}} \frac{1}{2\|\omega_{P,g}\|_\infty^2} \langle \pi_Q, (I - P)\omega_{P,g} \rangle^2 \tag{33}$$

We can now change the optimization variable from $g$ to $\omega$ as:

$$\sup_{\substack{g \in \mathbb{R}^m: \\ \langle \pi_P, g \rangle = 0}} \frac{1}{2\|\omega_{P,g}\|_\infty^2} \langle \pi_Q, (I - P)\omega_{P,g} \rangle^2 = \sup_{\substack{\omega \neq 0 \\ \langle \pi_P, \omega \rangle = 0}} \frac{\langle a, \omega \rangle^2}{2\|\omega\|_\infty^2} = \frac{1}{2} \left( \sup_{\substack{\|\omega\|_\infty = 1 \\ \langle \pi_P, \omega \rangle = 0}} \langle a, \omega \rangle \right)^2. \tag{34}$$

To justify the change of variable, observe that for every feasible $g \in \mathbb{R}^m$ (i.e., $\langle \pi_P, g \rangle = 0$), there exists a corresponding non-zero $\omega_{P,g}$ satisfying $\langle \pi_P, \omega_{P,g} \rangle = 0$. This implies that the value of the first optimization problem is at most that of the second. Conversely, for any non-zero $\omega$ such that $\langle \pi_P, \omega \rangle = 0$, the vector defined by $g = (I - P)\omega$ satisfies the constraint $\langle \pi_P, g \rangle = 0$. This ensures that the value of the second optimization problem is at most that of the first, proving the equality.

The inner supremum is a linear program:

$$\max_{\omega \in [-1,1]^m} \langle a, \omega \rangle \quad \text{s.t.} \quad \langle \pi_P, \omega \rangle = 0. \tag{35}$$

Constructing the Lagrangian with parameters $(\lambda_i)_{i \in [m]}, (\gamma_i)_{i \in [m]}, \mu$ such that $\lambda_i, \gamma_i \geq 0 \ \forall \ i \in [m], \mu \in \mathbb{R}$, we have

$$\mathcal{L}(\lambda, \gamma, \mu; \omega) = -\langle a, \omega \rangle + \sum_i \lambda_i(\omega_i - 1) + \sum_i \gamma_i(-\omega_i - 1) + \mu \langle \pi_P, \omega \rangle.$$

Observe that since the objective function as well as the inequality constraints are convex and the equality constraint is affine, strong duality is equivalent to the Kahn-Karush-Tucker (KKT) conditions (see for example, Vishnoi (2021, Theorem 5.12)). Thus, the optimizers $(\omega^*, \lambda^*, \gamma^*, \mu*)$ must satisfy the following KKT conditions:

1. *(Dual feasibility)* $\lambda_i^* \geq 0, \gamma_i^* \geq 0 \ \forall \ i \in [m]$.

2. *(Primal feasibility)* $\omega_i^* \in [-1, 1] \ \forall \ i \in [m], \quad \langle \pi_P, \omega \rangle = 0$.

3. *(Stationarity)* For each $i \in [m]$, we have: $\frac{\partial L}{\partial \omega_i} = 0 \implies a_i - \mu^* \pi_P(i) = \lambda_i^* - \gamma_i^*$.

4. *(Complementary slackness)* For each $i \in [m]$, we have: $\lambda_i^*(\omega_i^* - 1) = 0, \quad \gamma_i^*(-\omega^* - 1) = 0$. Observe that this implies at least one of $\lambda_i^*, \gamma_i^*$ must be 0.

We analyze the stationarity condition above case by case:

1. *Case* 1: $a_i - \mu^* \pi_P(i) > 0 \implies \lambda_i^* - \gamma_i^* > 0 \implies \lambda_i^* > 0 \implies \omega_i^* = 1$.

2. *Case* 2: $a_i - \mu^* \pi_P(i) < 0 \implies \lambda_i^* - \gamma_i^* < 0 \implies \gamma_i^* < 0 \implies \omega_i^* = -1$.

3. *Case* 3: $a_i - \mu^* \pi_P(i) = 0 \implies \lambda_i^* = \gamma_i^* = 0$.

Define a partitioning of the coordinates based on the sign of $a_i - \mu \pi_P(i)$:

$$\mathcal{S}^+(\mu) = \{i \in [m] : a_i - \mu \pi_P(i) > 0\}$$
$$\mathcal{S}^-(\mu) = \{i \in [m] : a_i - \mu \pi_P(i) < 0\}$$
$$\mathcal{S}^=(\mu) = \{i \in [m] : a_i - \mu \pi_P(i) = 0\}$$

Thus, the stationarity conditions dictate that for the optimal $\mu^*$: $\omega_i^* = 1$ for $i \in \mathcal{S}^+$, and $\omega_i^* = -1$ for $i \in \mathcal{S}^-$. To ensure primal feasibility ($\sum_i \pi_P(i)\omega_i^* = 0$), we must have:

$$\sum_{i \in \mathcal{S}^+} \pi_P(i) - \sum_{i \in \mathcal{S}^-} \pi_P(i) + \sum_{i \in \mathcal{S}^=} \pi_P(i)\omega_i^* = 0.$$

To find a $\mu^*$ that satisfies all constraints (thereby achieving the optimal value), consider the function:

$$C(\mu) = \sum_{i=1}^{m} \pi_P(i) \, \mathrm{sgn}(a_i - \mu \pi_P(i)).$$

This function has the following properties:

- $C(\mu)$ is a monotonically decreasing, piecewise constant function.

- The function steps at the points $\{a_i/\pi_P(i) : i \in [m]\}$. Let us arrange these points in strictly increasing order $\frac{a_{(1)}}{\pi_{P_{(1)}}} < \frac{a_{(2)}}{\pi_{P_{(2)}}} < \ldots$ (grouping identical ratios).

- The limits are $\lim_{\mu \to -\infty} C(\mu) = 1$ and $\lim_{\mu \to \infty} C(\mu) = -1$.

Because of its limits and monotonicity, we consider two scenarios for where $C(\mu)$ crosses zero:

1. *Case 1:* There exists an interval $\left[ \frac{a_{(j)}}{\pi_{P_{(j)}}}, \frac{a_{(j+1)}}{\pi_{P_{(j+1)}}} \right)$ where $C(\mu) = 0$. By choosing $\mu^*$ strictly inside this open interval, $\mathcal{S}^=(\mu^*) = \emptyset$. Because $C(\mu^*) = 0$, we have exactly $\sum_{i \in \mathcal{S}^+} \pi_P(i) = \sum_{i \in \mathcal{S}^-} \pi_P(i) = 1/2$. Setting $\omega_i^* = 1$ for $\mathcal{S}^+$ and $-1$ for $\mathcal{S}^-$ immediately satisfies primal feasibility.

2. *Case 2:* $C(\mu)$ jumps across zero at a specific step. That is, there exists an index $j$ such that $C(\mu) > 0$ for $\mu < \frac{a_{(j)}}{\pi_{P_{(j)}}}$ and $C(\mu) < 0$ for $\mu > \frac{a_{(j)}}{\pi_{P_{(j)}}}$. We set $\mu^* = \frac{a_{(j)}}{\pi_{P_{(j)}}}$. Here, $\mathcal{S}^=(\mu^*)$ contains $j$ (and potentially other coordinates with the same ratio). We choose $\omega_j^*$ to satisfy primal feasibility:

$$\pi_P(j)\omega_j^* = \sum_{i \in \mathcal{S}^-} \pi_P(i) - \sum_{i \in \mathcal{S}^+} \pi_P(i).$$

To prove $\omega_j^* \in [-1, 1]$, we must show that the right-hand difference lies in $[-\pi_P(j), \pi_P(j)]$. By construction of the jump over zero, the mass strictly above the threshold ($\mathcal{S}^+$) and strictly below the threshold ($\mathcal{S}^-$) are both strictly less than $1/2$. Since $\sum_{i \in \mathcal{S}^-} \pi_P(i) + \sum_{i \in \mathcal{S}^+} \pi_P(i) + \pi_P(j) = 1$, the absolute difference between the two subsets cannot exceed the remaining mass $\pi_P(j)$. Thus, $\omega_j^*$ is a valid assignment.

Having established the optimal solution $\omega^*$ for the inner linear program, we now map this solution back to the original unconstrained optimization problem over $g$.

Recall that to solve the supremum of the functional $\psi(g)$, we restricted our search space without loss of generality to the subspace $\{g \in \mathbb{R}^m : \langle \pi_P, g \rangle = 0\}$, which allowed us to utilize the exact reparameterization $g = (I - P)\omega$. Substituting our optimal $\omega^*$ into this mapping gives a base optimal direction $g^\star = (I - P)\omega^*$.

Furthermore, as observed at the beginning of the proof, the functional $\psi(g)$ is invariant to both shifting by constant vectors $(g \mapsto g + b\mathbf{1})$ and scaling by non-zero constants $(g \mapsto cg)$. Consequently, any vector that is a scaled and shifted version of $g^\star$ will achieve the exact same supremum.

Therefore, the full set of maximizers for the original optimization problem is exactly the set $\{c\,g^\star + b\mathbf{1} : c \neq 0, b \in \mathbb{R}\}$, which completes the proof.

$\square$

## E. Proofs for Section 5

**Lemma E.1.** *The set $P_\pi := \{P \in \mathcal{M} : \pi P = \pi\}$ is compact.*

*Proof of Lemma E.1.* Boundedness follows the fact that each element of $P_\pi$ is a stochastic matrix and thus has $\ell_{1,\infty}$ norm of 1. For closedness, consider the linear, continuous function $f : \mathbb{R}^{m \times m} \to \mathbb{R}^m, f(P) = \pi P$. The preimage of the closed set $\{\pi\}$ under this function, i.e. $f^{-1}(\pi)$ is a closed set. Closedness follows from observing that the space of stochastic matrices is closed, the intersection of closed sets is closed and $P_\pi = f^{-1}(\pi) \bigcap \mathcal{M}$. Compactness then follows from the Heine-Borel theorem (Kreyszig, 1978, Theorem 2.5-3).

$\square$

**Lemma E.2.** *The set $\mathcal{P}_L$ is compact under the standard topology.*

*Proof of Lemma E.2.* Observe that the set $\{\mu \in \mathbb{R}^{|\mathcal{S}| \times d} : \|\mu\|_{2,\infty} \leq \sqrt{d}\}$ is a compact set under the topology induced by the $L_{2,\infty}$. Since $\mathbb{R}^{|\mathcal{S}| \times d}$ is a finite dimensional normed space, all norms induce an equivalent topology. As a consequence, $\{\mu \in \mathbb{R}^{|\mathcal{S}| \times d} : \|\mu\|_{2,\infty} \leq \sqrt{d}\}$ is compact under the topology induced by the $\ell_{1,\infty}$ norm. $\mathcal{P}_L$ is the image of this set under the linear function given by $\mu \mapsto \Phi\mu^\top \Pi_M$, which is continuous. Since continuous functions map compact sets to compact sets, we have that $\mathcal{P}_L$ is compact. $\square$

## F. Technical Lemmas

**Lemma F.1.** *Let $a \geq 1$ and $b \geq 2a$. Then:*

$$x \leq a \log(x) + b \implies x \leq b + 2a \log(b).$$

*Proof.* We proceed by showing the contrapositive.

Let $f(x) := x - a \log x - b$. Then, $f'(x) = 1 - a/x$. Let $x_0 := b + 2a \log b$. Observe $x > x_0 \implies x > b > a \implies f'(x) > 0$. Therefore $f$ is strictly increasing $\forall x > x_0$.

Next, we show $b^2 > b + 2a \log b$.

$$b^2 - b = b(b-1) > b \log b > 2a \log b,$$

where the first inequality follows since $b > 1$, and second uses $b \geq 2a$.

Now, consider:

$$\begin{aligned}
f(x_0) &= b + 2a \log b - a \log(b + 2a \log b) - b \\
&= a(2 \log b - \log(b + 2a \log b)) \\
&= a \log \left( \frac{b^2}{b + 2a \log b} \right) \qquad\qquad (\because b^2 > b + 2a \log b)
\end{aligned}$$

$$> 0.$$

Since the function is strictly increasing, $f(x) > f(x_0) \; \forall \; x > x_0$ i.e. $x > a \log x + b \; \forall \; x > b + 2a \log b$.

$\square$

**Lemma F.2.** *Let* $d(x,y) := x \log\left(\frac{x}{y}\right) + (1-x) \log\left(\frac{1-x}{1-y}\right)$ *be the KL divergence between* $Ber(x)$ *and* $Ber(y)$ *for* $x, y \in (0,1)$. *Then:*

$$\lim_{\alpha, \beta \to 0} \frac{d(\beta, 1-\alpha)}{\log(1/\alpha)} = 1,$$

*Proof.* We expand the KL divergence as follows:

$$\frac{d(\beta, 1-\alpha)}{\log(1/\alpha)} = \frac{\beta \log\left(\frac{\beta}{1-\alpha}\right)}{\log(1/\alpha)} + \frac{(1-\beta) \log\left(\frac{1-\beta}{\alpha}\right)}{\log(1/\alpha)}$$

$$= \frac{\beta \log\left(\frac{\beta}{1-\alpha}\right)}{\log(1/\alpha)} + \frac{(1-\beta) \log(1-\beta)}{\log(1/\alpha)} + (1-\beta).$$

Taking the limit as $\alpha, \beta \to 0$, we see that the first two terms go to zero, while the last term goes to 1. $\square$

# G. Experiments

In this section, we provide details about our experimental methodology.

## G.1. Misspecification in MCMC

**Statistic vs time (fixed $\alpha$):** Here, we fix $\pi = [0.1, 0.1, 0.2, 0.2, 0.4]$ and $\alpha = 0.05$ and study the expected stopping times under two data generating instances.

1. $Q \notin \mathcal{P}_\pi$. We take the following matrix to generate the data:

$$Q_{\text{bad}} = \begin{bmatrix} 0.1 & 0.5 & 0.1 & 0.1 & 0.2 \\ 0.2 & 0.1 & 0.4 & 0.2 & 0.1 \\ 0.1 & 0.1 & 0.1 & 0.6 & 0.1 \\ 0.3 & 0.2 & 0.1 & 0.1 & 0.3 \\ 0.1 & 0.1 & 0.1 & 0.1 & 0.6. \end{bmatrix}$$

   We note that this matrix is positive and hence, specifies an ergodic Markov chain. Its corresponding stationary distribution is given as: $\pi_{\text{bad}} \approx [0.1574, 0.1825, 0.1548, 0.1956, 0.3097]$. We show the growth of the statistic and demonstrate how evidence accumulates against the null with time. We repeat the experiment over 100 runs and present the results in Figure 3.

2. $Q \in \mathcal{P}_\pi$. We take the following matrix to generate the data:

$$Q_{\text{good}} = \begin{bmatrix} 0.50 & 0.20 & 0.00 & 0.00 & 0.30 \\ 0.20 & 0.50 & 0.30 & 0.00 & 0.00 \\ 0.00 & 0.15 & 0.50 & 0.35 & 0.00 \\ 0.00 & 0.00 & 0.35 & 0.50 & 0.15 \\ 0.075 & 0.00 & 0.00 & 0.075 & 0.85 \end{bmatrix}.$$

   Naturally, the stationary distribution of this chain, $\pi_{\text{good}} = \pi$. We show in this case, the statistic does not grow and stays below the threshold. We repeat the experiment over 100 runs and present the results in Figure 4. We remark that, in our testing, the test did not stop erroneously.

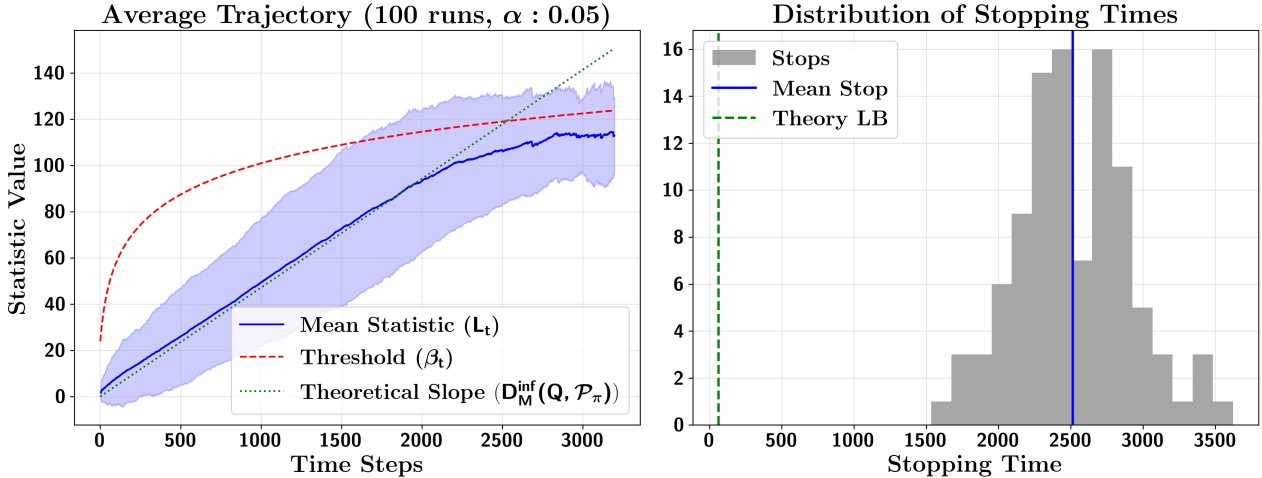

*Figure 3.* On the left side, we show the average trajectory (shaded: $\pm 3$ standard deviations) taken by the statistic across time. On the right side, we plot the histogram of stopping times across the same runs.

As a consequence, we estimate the probability of erroneously stopping under the null indirectly via importance sampling. We generate trajectories under $Q_{\mathrm{bad}}$, under which the test stops quickly and use the following a change of measure identity:

$$\mathbb{P}_{Q_{\mathrm{good}}}\left[\tau_\alpha < \infty\right] = \mathbb{E}_{Q_{\mathrm{bad}}}\left[\mathbb{I}\left\{\tau < \infty\right\}\frac{dQ_{\mathrm{good}}}{dQ_{\mathrm{bad}}}(\tau_\alpha)\right],$$

where $\frac{dQ_{\mathrm{good}}}{dQ_{\mathrm{bad}}}(\tau)$ denotes the likelihood ratio evaluated along the trajectory up to time $\tau_\alpha$. Using this estimator, we evaluated the false alarm probability to be $1.21 \times 10^{-30}$ (with a 95% confidence interval of $2.12 \times 10^{-30}$ over $n = 1000$ runs).

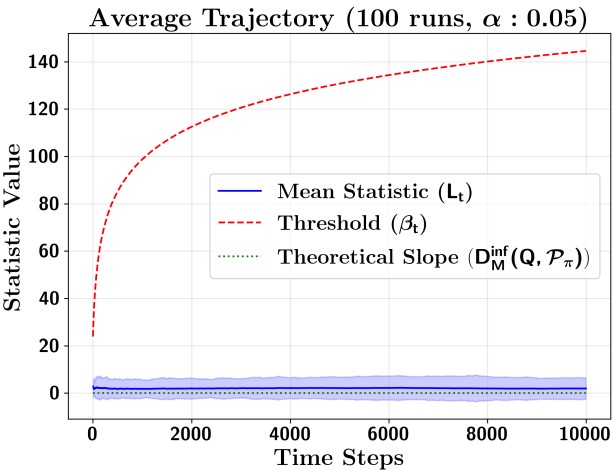

*Figure 4.* We show the average trajectory (shaded: $\pm 3$ standard deviations) taken by the statistic across time. We do not observe any rejections empirically and estimate the rejection probability indirectly via importance sampling to be $1.21 \times 10^{-30} \pm 2.12 \times 10^{-30}$ over $n = 1000$ runs.

### G.2. Linearity Testing in MDPs

Here, we provide details of the experimental setup implemented in Section 5.2.

We evaluate our testing framework on a discretized version of the classic MountainCar-v0 environment. The continuous state space, consisting of position $p \in [-1.2, 0.6]$ and velocity $v \in [-0.07, 0.07]$, is discretized into a uniform grid. We use

an $8 \times 8$ grid, resulting in a finite state space of size $S = 64$. The environment has a discrete action space of size $A = 3$ (accelerate left, no acceleration, accelerate right). The data stream $(X_0, A_0, X_1, A_1, \dots)$ is generated using a uniform random policy, where actions are selected uniformly at random $A_t \sim \text{Unif}(\mathcal{A})$ at each step.

We construct the feature map $\phi(s, a)$ using Radial Basis Functions (RBF). We treat the state-action pair index $k \in \{0, \dots, SA - 1\}$ as a scalar and define $d - 1$ Gaussian centers $\mathbf{c}_j$ evenly spaced across the domain. The feature vector is given by:

$$\phi(s, a) = \text{Normalize}\left(\left[1, \exp\left(-\frac{(k - \mathbf{c}_1)^2}{2\sigma^2}\right), \dots, \exp\left(-\frac{(k - \mathbf{c}_{d-1})^2}{2\sigma^2}\right)\right]^\top\right),$$

where a bias term of $1$ is included, and the resulting vector is normalized to sum to 1. We solve the convex optimization problem for the statistic using the cvxpy package (Diamond & Boyd, 2016; Agrawal et al., 2018). We test the linearity assumption across varying feature dimensions (ranks) $d \in \{3, 5, 7\}$ with the following choice of hyper-parameters:

*Table 1.* Hyperparameters for the Linear MDP Experiment on Discretized MountainCar.

| Parameter | Value | Description |
|---|---|---|
| $\alpha$ | 0.01 | Target Type-I error probability |
| $S$ | 64 | State space size ($8 \times 8$ discretization) |
| $A$ | 3 | Action space size |
| $d$ | $\{3, 5, 7\}$ | Tested feature dimensions (ranks) |
| Check Interval | 100 | Frequency of statistic evaluation (steps) |
| $T$ (Horizon) | $100,000$ | Maximum duration of the experiment |

## G.3. Experiments on Synthetic Data from a Parametric Family

We define a one-parameter exponential family of transition matrices $\{P_\theta : \theta \in \mathbb{R}\}$ derived from $P_0$ and a fixed feature vector $f \in \mathbb{R}^m$.

For each $\theta$, define the un-normalized matrix

$$\widetilde{P}_\theta(i, j) = P_0(i, j) \exp(\theta \cdot f_j),$$

Let $\rho_\theta$ denote the Perron–Frobenius eigenvalue of $\widetilde{P}_\theta$ and let $v_\theta$ be the corresponding positive right eigenvector normalized to satisfy $\sum_j v_\theta(j) = 1$. The normalized transition matrix is defined as

$$P_\theta(i, j) = \frac{\widetilde{P}_\theta(i, j) v_\theta(j)}{\rho_\theta v_\theta(i)}.$$

For our experiments, we consider a five-state Markov chain. We randomly generate a transition matrix $P_0$.

We then define null and alternative families as two disjoint parameter intervals:

$$\mathcal{P} = \{P_\theta : \theta \in [0.4, 0.8]\}, \qquad \mathcal{Q} = \{P_\theta : \theta \in [-0.8, -0.4]\}.$$

A single generating distribution $Q \in \mathcal{Q}$ is selected by fixing $\theta_Q = -0.6$ from the alternate family. We set $Q = P_{\theta_Q}$ and generate the new transition matrix as mentioned above.

All observed data sequences are generated from $Q$ in the reported experiments.

The feature vector is fixed as

$$f = (1, 1, 0, -1, -1).$$

We run each experiment for 100 epochs across varying values of $\alpha$ which is the Type I error level.

**State Size ($m$) vs Error Probability ($\alpha$):** Within the one-parameter exponential family framework above, we conduct experiments for varying state sizes and error levels. We run our experiments with Google Colab, Intel x86_64 Xeon CPU @2.20 GHz (1 core, 2 logical CPUs) and report the runtimes ($n = 100$ runs) in Tables 2 and 3 below.

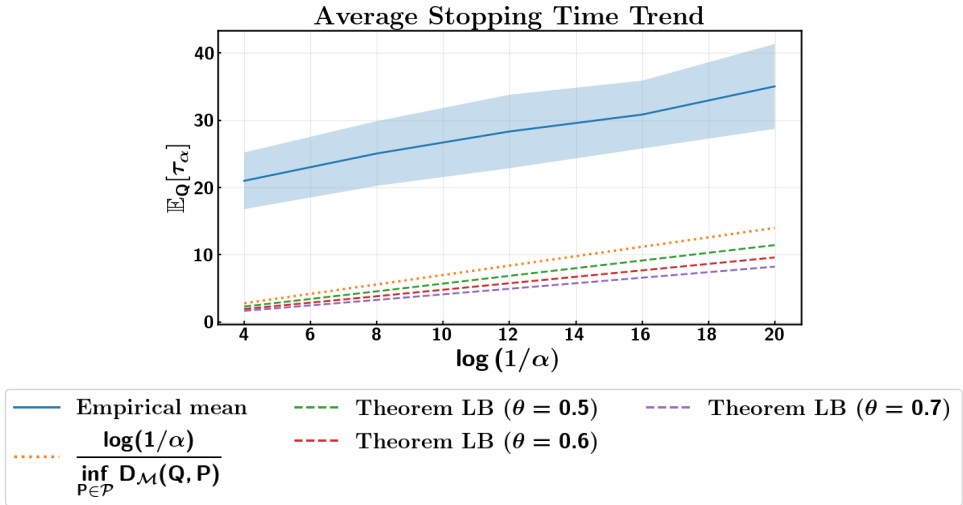

*Figure 5.* Expected Stopping Time as a function of $\log\left(\frac{1}{\alpha}\right)$ for the experimental setup described in Appendix G.3.

*Table 2.* Comparison of mean stopping times and runtimes for different values of $m$.

| m | $\alpha$ | Mean Stopping Time | Mean runtime/step (s) |
|---|---|---|---|
| 32 | 0.05 | $665.65 \pm 130.893$ | $0.0184 \pm 0.001$ |
| 64 | 0.05 | $2516.84 \pm 620.151$ | $0.0796 \pm 0.002$ |
| 128 | 0.05 | $9700.55 \pm 2181.039$ | $0.6890 \pm 0.121$ |

### G.4. Comparison with Baselines

Since there are currently no established baselines for sequential testing with composite null and composite alternative hypotheses, this experiment is not intended as a quantitative comparison of algorithmic efficiency or optimality. Instead, it is designed to illustrate the ability of the proposed method to adapt to varying levels of problem difficulty. To obtain a point of reference, we implement the procedure of Fields et al. (2025), which is designed for a simple null hypothesis and a composite alternative. We apply it in our setting by collapsing the null family to a singleton at $\theta = 0.2$, while retaining the composite alternative $\theta \in [-0.8, -0.4]$. All other aspects of the experimental setup are kept identical across methods.

*Table 3.* Comparison of mean stopping times and runtimes for different values of $\alpha$.

| m | $\alpha$ | Mean Stopping Time | Mean runtime/step (s) |
|---|---|---|---|
| 64 | 0.01 | $2864.83 \pm 565.180$ | $0.0692 \pm 0.004$ |
| 64 | 0.05 | $2516.84 \pm 620.151$ | $0.0796 \pm 0.002$ |
| 64 | 0.1 | $2370.36 \pm 717.654$ | $0.0773 \pm 0.002$ |

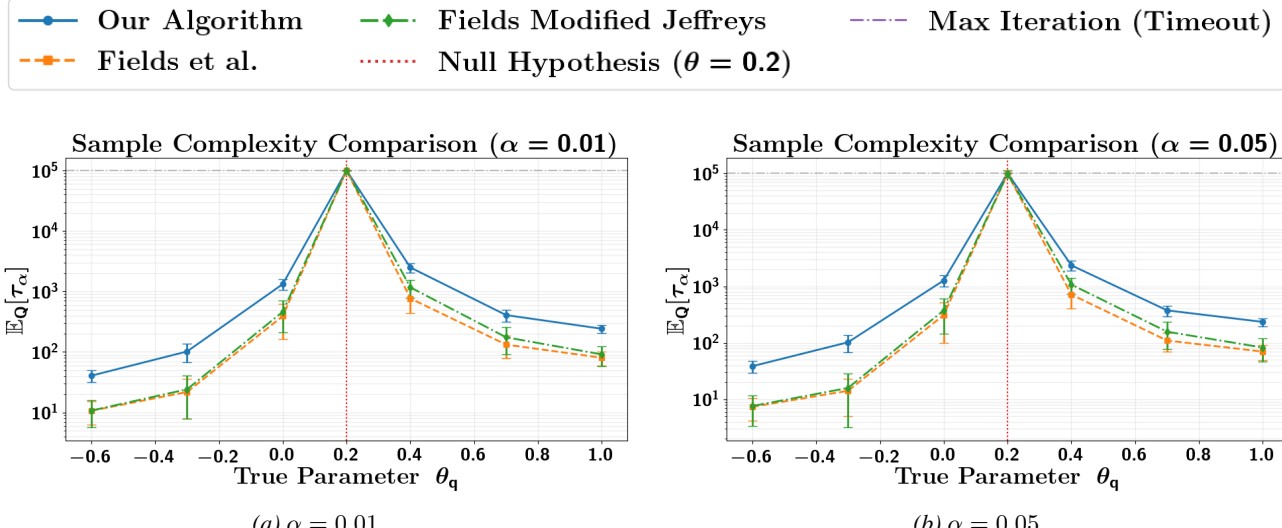

*(a)* $\alpha = 0.01$          *(b)* $\alpha = 0.05$

*Figure 6.* Expected stopping time as a function of $\theta \in [-0.8, -0.4]$ for the experimental setup described in Section G.4.

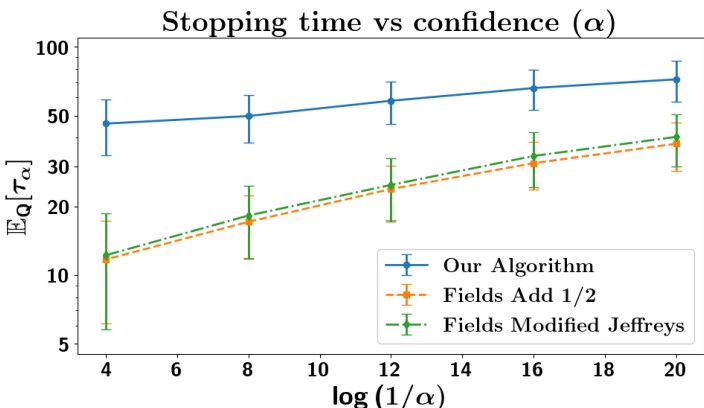

*Figure 7.* Expected Stopping Time as a function of $\log\left(\frac{1}{\alpha}\right)$ for a fixed $\theta = -0.6$ in the experimental setup described in G.4.

