# OpenReview forum: "Asymptotically Optimal Sequential Testing with Markovian Data"
_ICML.cc/2026/Conference — ICML 2026 regular_

### Official Review · Reviewer_zfST · 2026-02-16

**Soundness:** 4
**Presentation:** 4
**Significance:** 3
**Originality:** 3
**Overall Recommendation:** 5
**Confidence:** 5

**Summary:**

This work considers composite versus composite, one-sided, sequential hypothesis testing about the transition matrix of a Markov chain. A feasible test should have type-I error rate at most alpha under the null and power one under the alternative. A non-asymptotic lower bound on the expected sample size under any distribution in the alternative is established, which is of the form of log(1/alpha)/minimum KL divergence - a term due to the Markovianity. A test is proposed that is strictly feasible for any given alpha, and asymptotically optimal as alpha goes to zero. The proposed test is then applied to two concrete testing settings.

**Compliance With Llm Reviewing Policy:**

Affirmed.

**Key Questions For Authors:**

1. According to the proof of error control, using the martingale Mt and the threshold log(1/alpha) also leads to strict error control and is less conservative. I understand that the reason for using the current test statistic Lt instead of Mt is that Mt is computationally very hard. It is interesting to see their (and also underline(Lt)) difference in efficiency (expected sample size) in simulation studies, which reflects the statistical-computational tradeoff.
2. Although the error control is strict, it should be conservative in general. So it is ideal to see a figure comparing the nominal level and the actual level. Importance-sampling may be used here.

**Limitations:**

Yes.

**Strengths And Weaknesses:**

Soundness: This work establishes a complete asymptotic optimality theory. The proofs are basically sound and well-organized.
Presentation: The work is well-written, with clear definitions, theorems, proofs, figures, discussions, etc.
Significance: This work addresses the problem of sequential hypothesis testing about Markovian data, and the proposed algorithm is  computationally more efficient than the naive application of mixture/adaptive/generalized SPRT.
Originality: The methods and proof-techniques are basically standard, but applied in a very nice way.

---

> ### Author Rebuttal · Authors · 2026-03-31
>
> We thank the reviewer for their thoughtful review and positive assessment of our work. We are grateful for the time that the reviewer devoted to reading the paper, and for the particularly insightful comments.
>
> **Martingale $M_t$ versus $L_t$**
>
> The reviewer’s comment regarding the statistical–computational trade-off is both important and insightful, and we thank them for highlighting it.
>
> We agree that it would be very valuable to better understand the inherent statistical and computational trade-offs at play here. At the same time, we believe that carrying out such a comparison in a meaningful way is somewhat subtle. In particular, the practical performance of any surrogate depends critically on the tightness of the concentration bounds used in its construction. When these bounds are loose, empirical comparisons may not accurately reflect the underlying statistical–computational gap.
>
> For context, Eq. (21) in our paper shows that the additional term appearing in the threshold $\beta$ can be interpreted as the “worst-case regret” of $\log M_t$ relative to the weighted KL benchmark. This cost is closely tied to the choice of prior used to construct $M_t$. We believe that the present construction can likely be improved through alternative choices of prior, thereby reducing this regret term and bringing the threshold much closer to $\log(1/\alpha)$. Such improvements are known to be possible in the i.i.d. setting (Theorem 8, [1]).
>
> Importantly, such refinements primarily affect finite-$\alpha$ performance, since even the current thresholding of $L_t$, albeit conservative, already yields an asymptotically optimal test. Along related lines, the recent work of [2] develops data-dependent regret bounds that could potentially further improve stopping-time performance for a fixed $\alpha>0$. These approaches would still be based on thresholding $L_t$, but with thresholds that are potentially much closer, in a data-dependent way, to $\log(1/\alpha)$.
>
> We will revise the paper to explicitly discuss this trade-off and to highlight the importance of understanding it empirically. A systematic study of this question is, in our view, a very interesting direction for future work.
>
> **Nominal error vs actual error via importance sampling**
>
> We thank the reviewer for bringing this importance sampling technique to our attention. To estimate the actual type-I error probability $\mathbb{P}_P[\tau < \infty]$ for a null model $P \in \mathcal{P}$, we re-ran the experimental setup used to generate the trajectories in Figure 4 under an alternative model $Q \in \mathcal{Q}$, under which the test typically stops quickly. We then used the standard change-of-measure identity at the stopping time $\tau$:
>
> $$
> \mathbb{P}_P[\tau < \infty]  $$
>
> $$ = \mathbb{E}_{Q}\left[\frac{dP}{dQ}(\tau) \cdot \mathbf{1}\{\tau < \infty\}\right],
> $$
>
> where $\frac{dP}{dQ}(\tau)$ denotes the likelihood ratio evaluated along the trajectory up to time $\tau$. Using this estimator, we evaluated the empirical type-I error for the experimental setup shown in Figure 4 of the manuscript, where the nominal level is $\alpha = 0.05$. We found the actual empirical erroneous stopping probability to be approximately $3\times 10^{-7}$ (with a $95$ percent CI of $\pm 7\times 10^{-27}$).
>
> This confirms that the current test is quite conservative. As noted in the previous point, an interesting direction for future work is to reduce the gap between the nominal and actual error levels while preserving the theoretical guarantees.
>
> *References*
>
> [1] Orabona, F., & Jun, K. S. (2023). Tight concentrations and confidence sequences from the regret of universal portfolio. IEEE Transactions on Information Theory, 70(1), 436-455.
>
> [2] Agrawal, S., & Ramdas, A. (2025). Eventually LIL Regret: Almost Sure $\ln\ln T $ Regret for a sub-Gaussian Mixture on Unbounded Data. arXiv preprint arXiv:2512.12325.

---

### Official Review · Reviewer_ivjc · 2026-02-26

**Soundness:** 3
**Presentation:** 3
**Significance:** 3
**Originality:** 3
**Overall Recommendation:** 4
**Confidence:** 3

**Summary:**

The paper considers sequential testing for Markov chains under composite hypotheses. It establishes a tight, non-asymptotic lower bound and proposes an asymptotically optimal test. By leveraging the Poisson equation to control temporal dependence, the framework provides sharp guarantees, as demonstrated in MCMC diagnostics and linear MDP structural testing.

**Compliance With Llm Reviewing Policy:**

Affirmed.

**Key Questions For Authors:**

1. Regarding the tractable surrogate $L'_t$: what is the quantitative gap in sample efficiency compared to the optimal statistic $L_t$? Does this gap remain constant as $\alpha$ decreases?
2. How does the framework perform when the pseudo-spectral gap is very small? Does the non-asymptotic correction term become the dominant factor in those cases?
3. In the MDP experiment, could you clarify how the discretization level influences the linearity of the transition dynamics?

**Limitations:**

yes

**Strengths And Weaknesses:**

Strength:
1. Careful mathematical development with non-asymptotic lower bounds.
2. Applications to MCMC diagnostics and MDP structural testing address real problems.
3. First instance-dependent lower bounds for composite-null sequential testing with Markovian dependence.
Weakness
1. A limitation of the work is its reliance on finite state spaces. While the ergodic assumptions are clearly stated, the scaling of the complexity constants relative to the number of states can be more thoroughly analyzed.

---

> ### Author Rebuttal · Authors · 2026-03-31
>
> We thank the reviewer for their time and for the careful review of our work. We address the questions below.
>
> **Weakness 1: Finite state space**
>
> We agree that the finite-state nature of our current results should be stated more explicitly. We have revised the abstract and introduction to reflect the same. Please also refer to our response to reviewer **ypjK** (point about **'Large $ m $ setting'**) for discussion on the dependence on state size, computational complexity and possible relaxations.
>
> **Question 1: Sample efficiency of tractable surrogate**
>
> The testing problem that we study is quite general. Even in important special cases, characterizing the sample-complexity gap induced by tractable surrogate statistics remains open. For example, this question is still unresolved for best-policy identification in MDPs [1].
>
> The i.i.d. literature suggests that surrogate statistics based on Pinsker-type relaxations can still lead to order-optimal procedures, albeit with a suboptimal constant factor. Classical examples are the regret guarantees of UCB [2] vs. KL-UCB [3] and IMED [4]; see also the discussion in Section 3.4 of [5]. We conjecture that a similar phenomenon should hold in the Markovian setting as well. Establishing this, however, appears to require new techniques and is beyond the scope of the present paper.
>
> We will incorporate this discussion near the current discussion around Line 323–324 in the revised manuscript.
>
> **Question 2: Low spectral gap regime**
>
> When $ \gamma_{ps} \approx 0 $ (and note that $ \gamma_{ps}> 0$ under ergodicity), the chain mixes very slowly. As a consequence, for some initial distributions, the laws of $ \omega(X_0) $ and $ \omega(X_\tau) $ may differ significantly. This is important because our test is required to operate correctly for any initial distribution. We also note that minimax lower bounds for transition-matrix estimation scale inversely with the pseudo-spectral gap (Theorem 3.2, [6]).
>
> To interpret the role of the term $C_Q/\min_i \pi_i$, we note that for any fixed $Q$, this quantity is simply constant in $\alpha$. Hence, as $\alpha \to 0$, the dominant term is still $\log(1/\alpha)$, and there always exists a threshold below which the resulting lower bound is strictly positive and informative.
>
> By contrast, when $\gamma_{ps}=0$, the data-generating chain is not ergodic and its mixing time is infinite. It is therefore unclear how to extend our analysis to that regime. We refer the reviewer to our discussion of ergodicity in the response to reviewer **d1HY** (point **SW.1**).
>
> We also refer the reviewer to our response to reviewer **ypjK** (point **Q1**) for further clarification on the non-asymptotic lower bound, in particular the distinction between the two lower bounds in Theorem 3.3. and the fact that the second one arises from a worst-case bound on the sensitivity of the Poisson equation solution.
>
> **Question 3: Impact of discretization**
>
> A finer discretization provides a more faithful approximation of the underlying continuous environment and can make structural deviations, such as non-linearity, more visible. The trade-off is computational: finer discretization also leads to a much larger state space. This is precisely the large-$m$ regime discussed in our response to reviewer **ypjK** (point about **'Large $m$ setting'**), and we kindly point the reviewer there for a more detailed discussion.
>
> Once again, we thank the reviewer for their insightful comments. We hope the above clarifies the issues they raised.
>
> *References*
>
> [1] Marjani et al. (2021). Navigating to the best policy in markov decision processes. Advances in Neural Information Processing Systems
>
> [2] Auer, P., Cesa-Bianchi, N. & Fischer, P. Finite-time Analysis of the Multiarmed Bandit Problem. Machine Learning 47, 235–256 (2002)
>
> [3] Olivier Cappé. Aurélien Garivier. Odalric-Ambrym Maillard. Rémi Munos. Gilles Stoltz. "Kullback–Leibler upper confidence bounds for optimal sequential allocation." Ann. Statist. 41 (3) 1516 - 1541, June 2013.
>
> [4] Honda, J., & Takemura, A. (2010). An Asymptotically Optimal Bandit Algorithm for Bounded Support Models. Annual Conference Computational Learning Theory.
>
> [5]  Agrawal, S., Juneja, S.K. &amp; Koolen, W.M.. (2021). Regret Minimization in Heavy-Tailed Bandits. Proceedings of Thirty Fourth Conference on Learning Theory, in Proceedings of Machine Learning Research.
>
> [6] Geoffrey Wolfer. Aryeh Kontorovich. "Statistical estimation of ergodic Markov chain kernel over discrete state space." Bernoulli 27 (1) 532 - 553, February 2021.

---

> > ### Author Rebuttal · Reviewer_ivjc · 2026-04-03
> >
> > Thank you. My concerns are addressed.

---

### Official Review · Reviewer_ypjK · 2026-03-10

**Soundness:** 3
**Presentation:** 2
**Significance:** 1
**Originality:** 3
**Overall Recommendation:** 3
**Confidence:** 4

**Summary:**

In this paper, the authors develop a sequential hypothesis testing procedure to test for a discrete Markovian data on whether a transition matrix $P$ of a Markov chain lies in a null class $\mathcal{P}$ or not. The authors derive a nonasymptotic lower bound (Thm 3.3) of the stopping time of $\alpha$ correct power 1 sequential test in terms of $D_{\mathcal{M}}^{inf}(Q,P)$, the infimum of stationary weighted KL divergence where infimum is taken over the null class $P\in\mathcal{P}$. Motivated by this lower bound, the authors develop a corresponding testing procedure (Algorithm 1). Authors also show that this lower bound is asymptotically optimal (Thm 4.1 combined with an equation in "proof sketch" page 4 right), as well as for the two-sided testing version in Theorem 4.3. Due to the infimum involved in Algorithm 1 line 16 of calculation of $L_t$, authors discuss an approximation scheme.

Authors present two application examples of the proposed sequential hypothesis testing procedure, (1) one for testing whether MCMC samples are drawn from a correct target distribution, and (2) one for testing whether the assumption of linear Markov decision processes is appropriate or not for a given sequence of state-action pairs. Authors perform a small empirical validation for each application example, (1) testing a misspecified Markov transition kernel under the setting of $m=5$ (Appendix H.1) and (2) linearity testing in MDP under the setting of $m=64\times 3 =192$.

**Compliance With Llm Reviewing Policy:**

Affirmed.

**Final Justification:**

My main concern is the practical usefulness of the non-asymptotic theory result presented in the manuscript (although the bounds are asymptotically optimal). My concerns are largely unresolved since 1) the examples (testing MCMC, linear MDP) should be demonstrated under large $m$ settings in order to be practically important in my view, but all examples are based on small $m$ settings, 2) the proposed algorithm could suffer from increasing dimensionality of $m$, both in theory (potentially vacuous lower bound on stopping time) and practice (significant slowdown of running time of the algorithm as $m$ grows). For theory, since $\min\pi_i$ is at most $1/m$ and $C_Q=2$ in ideal settings (i.i.d. samples), I am still concerned that the key nonasymptotic theory result presented in the manuscript, Theorem 3.3, could remain vacuous since $\alpha$ should be less than $\exp(-4mD^{inf}_M(Q,P))$ under such ideal setting in order to the lower bound to be strictly positive, and requiring $D^{inf}_M(Q,P)$ to shrink faster than $1/m$ to offest $m$ term seems a very restrictive condition in order to obtain non-vacuous lower bound independent of $m$. Since one should fix the type 1 error $\alpha$ before running the sequential testing, not adapting $\alpha$ based on the number of state spaces $m$, I believe it is important to ensure non-asymptotic bounds informative and relevant in practice. Thus, I keep my recommendation as is

**Key Questions For Authors:**

While authors emphasize the nonasymptotic bound result in Theorem 3.3, it is unclear whether the bound described in the expression (3) actually give a practically useful lower bound or not (see also Fig 3(right)). For instance, there is little insight and discussion given on at what circumstances the right-hand side of  expression (3) is actually nonzero, ideally with illustration with concrete examples, and what that implies for the condition of $\min_i \pi_i$ and $C_Q$ in order to the nonasymptotic lower bound to be useful (since arbitarily small $\min_i \pi_i$ would always make RHS of the  expression (3) zero.)

**Limitations:**

There is no potential negative societal impact of the work

**Strengths And Weaknesses:**

The manuscript is generally well written, and the authors provide a detailed description on proof strategy of key theoretical results under a minimal assumption on the ergodicity of the Markov chain. It is important to clarify that this work focuses on a finite state space setting, which would be better clarified in the title or abstract. Aside from the theoretical contribution, which I acknowledge, I have two main concerns about the significance of the result for practical settings and the feasibility of the algorithm in such settings.

**Significance**. The authors present two examples where the proposed sequential testing procedure can be deployed: (1) testing MCMC convergence (2) testing linear MDP assumption. In practice, these two examples are only relevant for very large $m$ settings but not for small to moderate $m$. For example, let's say $m$ is roughly several thousand. Then, (1) running MCMC to sample from a target distribution with $m$ states is meaningless because one can directly sample from a target distribution, thus the problem of testing MCMC becomes nonexistent in practice, (2) one could simply adopt a tabular MDP if $m$ is moderate, which does not require assumptions in linear MDP, by representing a value function by a table where the whole table can be stored in memory, thus the problem of testing whether linearity assumption is appropriate or not becomes pointless. In other words, it is unclear whether the proposed sequential testing procedure could be applied to a situation where $m$ is exponentially large. Such examples include a variable selection problem where $m=2^p$ (p is a number of variable) where running MCMC on a discrete state space is necessary (since direct sampling is impossible) and testing MCMC convergence becomes relevant; similarly for linear MDP where the key motivation of linear MDP is aimed for very large $m$ settings, where tabular MDP becomes infeasible.


**Feasibility for large $m$**. The proposed sequential testing procedure described in Algorithm 1 involves iterative calculation of the test statistic $L_t$ in line 16, which involves taking an infimum over $\mathcal{P}$ of an empirical stationary weighted KL divergence. As discussed in Section 4.1, the space $\mathcal{P}$ must be convex in order to at least hope for finding such an optimum, and the authors suggest a surrogate statistic of $L_t$ in eq. 6 for a computationally tractable alternative. However, related to the previous comment, it is unclear what the complexity of the algorithm is in terms of $m$ and how big $m$ could possibly allowed in order to calculate $L_t$ or its surrogate in a reasonable amount of time.

---

> ### Author Rebuttal · Authors · 2026-03-31
>
> We thank the reviewer for the careful and thoughtful review, for recognizing the paper’s theoretical contributions, and for the concrete suggestions.
>
> **Finite-state:** We have now clarified this in the abstract and introduction of the revised manuscript.
>
> **Large $m$ settings** We will revise the paper (Sec. 4) to discuss the following points on computational effects of large $m$ explicitly.
>
> *(a) Tractable parametric form:* The computation of $L_t$ becomes much more tractable when $\mathcal P$ has low-dimensional parametric form. E.g., if $\mathcal P=\{P_\theta:\theta\in\Theta \subset\mathbb R^d\}$ is an exponential family with
>
> $$ \log P_{\theta}(x,y) = \langle \theta, T(x,y)\rangle - A_x(\theta) + \log h(x,y), $$
>
> then Line 16 becomes
>
> $$ L_t  = C_t +  \inf_{\theta\in\Theta} \left( \sum_x N_x A_x(\theta) - \Big\langle \theta,\sum_{x,y}N_{x,y}T(x,y)\Big\rangle \right),
> $$
>
> where $C_t$ depends only on the empirical counts. The optimization is then over the parameter $\theta$ of dimension $d$, rather than over $O(m^2)$ many transition probabilities [2]. In many such models, the optimization problem admits closed-form updates or is efficiently solvable using convex optimization routines. In structured large-$m$ settings, the complexity is governed by model dimension $d$ rather than ambient tabular size $m$.
>
> *(b) General computation:*  Computing the statistic involves 2 main steps: (a) estimating the transition matrix and evaluating the objective for a fixed $P\in\mathcal P$; (b) optimizing over $P\in\mathcal P$. In the tabular setting, (a) costs at most $O(m^2)$, while (b) depends on the geometry of $\mathcal P$. Thus, the bottleneck depends not only on $m$, but also on the structural complexity of the null class. In some cases, this can be efficient-- [1] show that an approximate statistic can be computed using $O(m)$ projected-gradient steps for policy testing in MDPs.
>
> *\(c\) Functional approximation:* The main strength of our framework is its generality. But for very large $m$ without additional structure (as in (b)), a natural approach is to combine our test with
>
> (i) state-space reduction/aggregation [6, 7]: first learn a compressed representation during an initial burn-in period, and then run our sequential test on the aggregated process;
>
> (ii) function approximators [3, 4, 5]: use regressors (e.g. neural nets, kernels etc.) to learn a PAC model of the transition matrix on-the-go (as done in model-based RL) and use it further to conduct the sequential test while re-calibrating the test for the learning error. We will include this in Sec 6.
>
> **(Q1.)** *(a) Non-negativity as $\alpha\to0$:* For any fixed $Q$, $C_Q/\min_i\pi_i$ is constant in $\alpha$. Hence, the lower bound (LB) is strictly positive for all sufficiently small $\alpha$, since $\log(1/\alpha)\to\infty$ as $\alpha\to0$.
>
> Fig. 3 (right) compares our LB and upper bounds. The visible gap is expected at the plotted values of $\alpha$: the asymptotic regime has not yet fully set in, and the empirical curve also reflects stochastic variability. Our theory guarantees only that this gap vanishes as $\alpha\to 0$.
>
> *(b) Role of $C_Q/\min_i\pi_i$:* We note that the proof of Theorem 3.3 first yields
>
> $$
> \sup_{P\in \mathcal{P}} \left( \log(1/\alpha) / D_M(Q, {P}) - E_Q [ \omega (X_0) - \omega (X_\tau) ] / D_M(Q, P) \right),
> $$
>
> where $\omega$ depends on $P$. In the i.i.d. setting, the numerator of the second term is zero for every $P\in\mathcal P$, recovering the known tight i.i.d. LB [8]. The same term also vanishes in the Markov setting when the initial distribution is the stationary distribution.
>
> To obtain a simpler $P$ indpendent bound, we upper bound $\mathbb E[\omega(X_0)-\omega(X_\tau)]/D_M(Q,P)$ uniformly yielding $C_Q/\min_i\pi_i$ originating from a triangle-inequality. While this makes the resulting non-asymptotic bound looser than the first one, both are asymptotically tight as $\alpha\to0$. Our looser bound still improves on existing non-asymptotic LB, which are off by a multiplicative factor and remain suboptimal even as $\alpha\to0$ (Section A.4). We will include this in the revision.
>
> *References:*
>
> [1] Ariu, K. et al. (2025). Policy Testing in Markov Decision Processes.
>
> [2] Moulos, V. Optimal best Markovian arm identification with fixed confidence. NeurIPS, 2019.
>
> [3] Buşoniu, L. et al. (2011). Approximate reinforcement learning: An overview. IEEE ADPRL (2011).
>
> [4] Wang, R. et al. (2020). What are the statistical limits of offline RL with linear function approximation?
>
> [5] Wang, R. et al. (2020). Reinforcement learning with general value function approximation: Provably efficient approach via bounded eluder dimension. NeurIPS.
>
> [6] Duan, Y. et al. (2019). State aggregation learning from Markov transition data. NeurIPS.
>
> [7] Zhang, A. et al. "Spectral State Compression of Markov Processes," IEEE Trans. on IT, 2020.
>
> [8] Agrawal, S. et al. (2025). On Stopping Times of Power-one Sequential Tests: Tight Lower and Upper Bounds.

---

> > ### Author Rebuttal · Reviewer_ypjK · 2026-04-03
> >
> > I acknowledge the authors' rebuttal. I have several follow-up questions:
> >
> > > it is unclear what the complexity of the algorithm is in terms of  $m$ and how big $m$ could possibly allowed in order to calculate or its surrogate in a reasonable amount of time.
> >
> > 0) I would appreciate it if authors could discuss my previous point (quoted above), e.g. wall-clock running time divided by time steps $t$ of Algorithm 1 as a function of $m$ and $\alpha$.
> >
> > 1) For (a) in author rebuttal, under the transition matrix $P_\theta$ with an exponential family form (I set $h(x,y)=1$ for simplicity which does not affect optimization over $\theta$ anyways), it seems the calculation of $\sum_x N_x A_x(\theta) = \sum_x N_x \log \sum_y \exp \langle \theta, T(x,y)\rangle$ still requires double summation over all finite states for a given $\theta$, thus the cost of evaluating objective function remains at $O(m^2)$.
> >
> > 2) For (c), when $m$ is very large, the authors suggest two approaches at a very high abstract level: state-space reduction/aggregation and function approximators. Could you explain in detail, with concrete examples, how the two approaches can be potentially combined for (1) testing misspecification in MCMC samplers and (2) testing linearity of MDPs with very large $m$ settings? For example, what is the "aggregated process" in those two application examples? How are states combined? How can testing on the aggregated process generalize to the testing of the original process?
> >
> > 3) Strict positiveness (not nonnegativity) of RHS of expression (3) as $\alpha\to 0$: this asymptotic behavior is obvious, as the authors already showed asymptotic optimality in Theorem 4.1. Given the author's highlighted contribution in the abstract "*We establish a **tight non-asymptotic** instance-dependent lower bound on the expected stopping time of any valid sequential test under the alternative*", my question is that for some fixed small $\alpha>0$ (non-asymptotic setting), under what situaton we could expect the lower bound (RHS of expression (3)) being actually nonzero & useful. If the "tightness" claim is based on the $\alpha \to 0$ asymptotics, the phrasing "tight non-asymptotic instance-dependent lower bound" is misleading, in my opinion.

---

> > > ### Author Response · Authors · 2026-04-08
> > >
> > > 0. **Runtime w.r.t. $m$ and $\alpha$**: We run our experiments with Google Colab, Intel x86_64 Xeon CPU @ 2.20 GHz (1 core, 2 logical CPUs) and report the runtimes as per the setup of Exp. H.3.
> > > - (a) Fix $\alpha$ \& vary $m$: observe that mean runtime/step scales polynomially in $m$.
> > >
> > > | m | α | Mean Stopping Time | Mean runtime/step (s)|
> > > | -------- | -------- | -------- | -------- |
> > > | 32 | 0.05 | $1505.8 \pm 90.023$ | 0.0174 |
> > > | 64 | 0.05 | $1457.8 \pm 40.535$ | 0.0777 |
> > > | 128 | 0.05 | $8193.4 \pm 142.548$ | 0.8338 |
> > >
> > > - (b) Fix $m$ \& vary $\alpha$: $\alpha$ only affects the stopping time but not the runtime per step, which is theoretically consistent.
> > >
> > > | m |  α | Mean Stopping Time | Mean runtime/step (s)|
> > > | -------- | -------- | -------- | -------- |
> > > | 64 | 0.01 | $1471.5 \pm 23.325$ | 0.0776 |
> > > | 64 | 0.05 | $1457.8 \pm 40.535$ | 0.0777 |
> > > | 64 | 0.1 | $465.2 \pm 27.194$ | 0.0778 |
> > >
> > > 1. **Exponential families of TPMs (transition probability matrices) and runtime:** The computation in Alg 1 involves two steps.
> > > (a) Evaluation of $L_t$ for a candidate $P$ and optimization over all $P \in \mathcal P$. In the generic tabular case, the evaluation for a candidate $P$ costs $O(m^2)$ independent of any structural assumptions.
> > > (b) Optimization over $P \in \mathcal P$ that operates with $O(m^2)$ variables in the generic tabular case but can be reduced under structural constraints on $\mathcal P$. For example, a single parameter exponential family (SPEF) of TPMs [1--3] is
> > > $$
> > > P_\theta(x,y)=\frac{R(x,y)e^{\theta f(y)}v_\theta(y)}{\rho_\theta\,v_\theta(x)}.
> > > $$
> > > Here, $R$ is a reference irreducible TPM, $f$ is the scalar statistic, and $(\rho_\theta,v_\theta)$ is Perron-Frobenius eigenpair of the matrix $M_\theta(x,y)=R(x,y)e^{\theta f(y)}$. For SPEF, the computation of $L_t$ (in Alg 1) simplifies to
> > > $$
> > > L_t= C_t+\inf_{\theta\in\Theta}\{t\log \rho_\theta -\theta\sum_{s=0}^{t-1}f(X_{s+1}) +\log v_\theta(X_0)-\log v_\theta(X_t)\},
> > > $$
> > > where $C_t=\sum_{x} N_x D_{KL}(\widehat Q_t(x,\cdot) \| R(x, \cdot))$ does not depend on $\theta$, and can be evaluated in $O(m)$ time in an online fashion. Thus, the row-wise KL objective collapses to three terms only: a scalar empirical sufficient statistic $\sum_{s=0}^{t-1}f(X_{s+1})$ (*updates in $O(1)$ in online fashion*), the log-eigenvalue $\log\rho_\theta,$ and a boundary correction involving $X_0$ and $X_t$ (*requires $O(m^2)$ for evaluation*).
> > > Thus, the computational gain is that the optimization problem for SPEFs reduces to dimension $1$ (e.g. line search over $\theta$) rather than an $O(m^2)$ optimization space. Thus, it is this reduction in optimization complexity, rather than the cost of a single objective evaluation yielding a computational gain.
> > >
> > > 2. **State-space reduction:**
> > > - **Trade-offs:** Consider $f:[m]\to [p]$ ($p\ll m$). If the partition induced by $f$ on $[m]$ is strongly lumpable [4], then if $(X_i) \sim P$, $(f(X_i))$ is Markov and has transition probabilities $\tilde{P}(a, b) = \sum_{j\in f^{-1}(b)}P_{i, j} := f(P)(a,b)$ for any $i\in f^{-1}(a), a, b\in [p]$.
> > > We can rewrite the original testing problem in the smaller space $[p]$, and the projected null and alternate: $\tilde{\mathcal{P}} = \{f(P):P\in \mathcal{P}\}$ and $\tilde{\mathcal{Q}} = \{f(Q):Q\in \mathcal{Q}\}$.  Compared to the original test, we have: a) **larger sample complexity** (stopping time) because of data processing inequality on ${D_M^{\inf}(Q, \mathcal{P})}$; b) **reduced per iteration cost**.
> > > *Thus, we are optimal in the reduced space but we can test upto equivalence classes induced by $f$ in the original space*.
> > > - **Appliction: testing misspecification in MCMC**. The original target distribution $\pi$ pushes forward to a target distribution $\tilde{\pi}$ over $[p]$, where $\tilde{\pi}(y) = \sum_{x \in f^{-1}(y)} \pi(x)$. If the original MC $Q$ has $\pi$ as its the stationary distribution, the aggregated MC $\tilde{Q}$ must have $\tilde{\pi}$ as its stationary distribution. If we reject on the smaller space, we are guaranteed that the null was false in the original space as well. Designing optimal ways to aggregate states and test the hypotheses is a future work (ref. Line 433, right).
> > >
> > > 3. **Strict positivity of lower bound**: RHS in Eq. (3) is strictly positive whenever $\alpha \in (0, \exp(-{2C_Q D_M^{\inf}(Q, \mathcal{P})}/\pi_*))$. We clarify that our contribution is the first non-asymptotic lower bound, which is provably tight in the limit $\alpha\to0$, as stated in introduction (Line 58-68). We will update the abstract with "We establish a non-asymptotic instance-dependent lower bound...which is asymptotically tight" to accurately reflect the theorem.
> > >
> > > [1] A convexity property in the theory of random variables defined on a finite Markov chain,1961
> > > [2] Optimal best Markovian arm identification with fixed confidence,2019
> > > [3] The exponential family of Markov chains and its information geometry,2017
> > > [4] A dual eigenvector condition for strong lumpability of Markov chains,2007

---

### Official Review · Reviewer_d1HY · 2026-03-11

**Soundness:** 4
**Presentation:** 4
**Significance:** 4
**Originality:** 3
**Overall Recommendation:** 5
**Confidence:** 4

**Summary:**

The submission studies the expected stopping time  of  sequential tests of a composite alternative hypothesis on ergodic Markov chain transition matrices versus a  null hypothesis consisting of   a disjoint set ofergodic  transition matrices.

A novel inequality is provided for the expectation of the stopping time  for an alpha-correct, power one sequential test  (Thm, 3.3). An algorithm is given for computing  such a  test.

**Compliance With Llm Reviewing Policy:**

Affirmed.

**Key Questions For Authors:**

Q1: Can the authors add   a discussion/empirical analysis  of  sequential testing  and  reenforcement learning?
Q2: Can  the proofs in Appendix E  be removed without any substantial clarity being lost?

**Limitations:**

The authors  have not  discussed the  potential negative societal impact of their work, but it is hard to give any suggestions on how to this in context. There are probably no immediate negative societal impact of this work.

**Strengths And Weaknesses:**

The paper   consists   mainly of theoretical results with correct  the proofs.   The theorems  require the  assumption that t the data-generating  Markov chain is ergodic, which is not mathematically reasonable.

The submission is  clearly written and well structured.   The work  does seem to position itself  properly in the context of prior  literature.  However, the literature  on stopping times and  sequential testing is very extensive and has been accumulating since 1945.

A weakness of the submission is what happens in its empirical  testing.  The authors study a Markovian decision  problem and test its linearity, which is defined in terms of  the transition matrices (see  Definition 5.2.).

This submission is  partially  evaluated   w.r.t.  understanding, capabilities, or practice in machine learning. There is no explicit connection established to machine learning in this work,  with the exception of  Markov chain Monte Carlo.
This is an unfortunate miss, as  Markovian decision problems are   the workhorse (?) of reinforcement learning. This    is a machine learning theory  where an autonomous agent learns to make optimal decisions by interacting with a dynamic environment through trial and error.

---

> ### Author Rebuttal · Authors · 2026-03-31
>
> We thank the reviewer for their time, their positive assessment of the theoretical contributions of our work, and their constructive suggestions for improving the submission. We address the questions below.
>
> **(SW1.)** We first note that the ergodicity assumption is standard in several existing works on learning and testing with Markovian data [1,2,3,4].
>
> While the assumption may appear strong at first, note that any stochastic matrix can be approximated arbitrarily well by an ergodic one (Line 116-120, right).
>
> More importantly, ergodicity guarantees convergence of the chain, which is central to our analysis. Specifically, we use ergodicity in two key places:
> (a) to ensure existence of a solution to the Poisson equation (PE) and to control its relative magnitude, which is one of the main technical ingredients behind our non-asymptotic lower bound; and
> (b) to invoke concentration results for estimating the transition matrix and the test statistic.
>
> We note here that it may be possible to weaken this assumption. Existence of PE's solutions and concentration phenomena might be accessible under irreducibility alone. However, it is presently unclear whether the type of bound that we require on the relative magnitude of the PE solution can _also_ be obtained in that weaker regime. Thus, we view this as an interesting future work.
>
> We will revise the manuscript to better explain the role of ergodicity and  expand the discussion following Assump. 2.1.
>
> Due to space limitations, we focused on prior work most directly relevant to our setting. While Section 1.2 was intended to cover this, we agree that a broader body of work in the i.i.d. setting can be acknowledged more explicitly. We will therefore cite the overview in Section 2 of [5] and expand the appendix to highlight related directions beyond the i.i.d. setting, including connections to property testing [6].
>
> **(SW3. and Q1.)** We agree that MDPs are fundamental to modern RL, and thus, we included the related work elaborating the connections with "Multi-armed bandits and RL" (Sec 1.2) and the linear MDP structural testing application (Sec 5.2).
>
> Given an MDP, any fixed policy induces a Markov chain on the state-action space. Our framework is general enough to test properties of this induced Markov chain, including linear structure and other structural assumptions such as low-rankness.
>
> The formulation and empirical evaluation in Section 5.2 focus on linearity because of its importance in RL, particularly in safety-critical settings. Since the introduction of linear MDPs, several works have relied on ergodicity-type assumptions to establish theoretical safety guarantees [7]. For such guarantees to be meaningful in practice, one must first assess whether the assumed linear structure is actually valid. To the best of our knowledge, our work provides the first principled sequential testing framework for this purpose.
>
> Furthermore, *we note that our test can be deployed parallely with an RL agent* since it only requires the trajectory of transitions generated by a fixed policy, it does not interfere with the RL agent's primary data collection or learning procedure.
>
> In RL, ergodicity is often used in finite-state settings to establish convergence of stochastic approximation algorithms [8,9], and in the episodic setting, it is induced through episodic resets [10]. We will make these connections more explicit in the revised version in Sec. 5.2.
>
> **(Q2.)** We included the proofs in Appendix E mainly for completeness and to provide concrete examples showing that it suffices to verify compactness of the null hypothesis class in order to apply our framework.
>
> Once again, we thank the reviewer for their time and insightful comments. We hope our responses have addressed their concerns clearly.
>
> *References:*
>
> [1] Nagaraj, D. et al. (2020). Least squares regression with markovian data: Fundamental limits and algorithms. NeurIPS.
>
> [2] Ziemann, I., & Tu, S. (2022). Learning with little mixing. NeurIPS.
>
> [3] P. N. Karthik et al. "Optimal Best Arm Identification with Fixed Confidence in Restless Bandits," IEEE Trans. on Info. Theory, 2024.
>
> [4] Fields, G., Javidi, T., & Shekhar, S. (2025). Sequential one-sided hypothesis testing of Markov chains.
>
> [5] Agrawal, S., & Ramdas, A. (2025). On Stopping Times of Power-one Sequential Tests: Tight Lower and Upper Bounds.
>
> [6] Batu, T. et al. (2013). Testing closeness of discrete distributions. Journal of the ACM.
>
> [7] Amani, S., Thrampoulidis, C., & Yang, L. (2021). Safe reinforcement learning with linear function approximation. ICML.
>
> [8] Srikant, R., & Ying, L. (2019). Finite-time error bounds for linear stochastic approximation andtd learning. COLT.
>
> [9] Chen, Z., et al. (2025). A non-asymptotic theory of seminorm lyapunov stability: From deterministic to stochastic iterative algorithms.
>
> [10] Dann, C., Lattimore, T., & Brunskill, E. (2017). Unifying PAC and regret: Uniform PAC bounds for episodic reinforcement learning. NeurIPS.

---

> > ### Author Rebuttal · Reviewer_d1HY · 2026-04-03
> >
> > I appreciate the authors'  comment on ergodicity.  I regret my typing error:  I wrote
> >
> > *which is not mathematically reasonable'
> >
> >
> > when I, of course, meant
> >
> >
> > 'which is   mathematically reasonable'.

---

### Decision · Program_Chairs · 2026-04-30

**Decision:**

Accept (regular)

**Comment:**

This paper studies the one-sided correct sequential hypothesis testing for data generated by an ergodic Markov chain. For a small error probability \alpha, this paper establishes a lower bound taking the form \log(1/\alpha)/(inf KL) for any sequential testing procedures, and constructs an algorithm that attains an upper bound with the same leading term. The authors present two applications of the proposed sequential hypothesis testing procedure: (1) testing whether MCMC samples are drawn from a correct target distribution, and (2) testing whether the assumption of linear Markov decision processes is appropriate or not for a given sequence of state-action pairs.

The reviewers generally appreciate that this paper is well written, states clear theoretical results with sound proofs, and conducts sound experiments. A criticism proposed by one reviewer is that the leading term only dominates when \alpha is very small (or the state space m is not too large). While this is a valid criticism, such lower-order terms are prevalent in the literature of instance-dependent bounds, so I do not view it as a major weakness. In addition, in the rebuttal the authors included experiments on large m, which also alleviates this concern. Therefore, I am happy to recommend acceptance, and suggest the authors to include theory/empirical discussions on large m.